# Beyond Linear Processing: Dendritic Bilinear Integration in Spiking Neural Networks

Jingyang Ma[a,b], Chongming Liu[a,b], Songting Li[a,b,c,*], and Douglas Zhou[a,b,c,d,e,*]

[a]School of Mathematical Sciences, Shanghai Jiao Tong University, Shanghai, China
[b]Institute of Natural Sciences, Shanghai Jiao Tong University, Shanghai, China
[c]Ministry of Education Key Laboratory of Scientific and Engineering Computing, Shanghai Jiao Tong University, Shanghai, China
[d]Shanghai Frontier Science Center of Modern Analysis, Shanghai Jiao Tong University, Shanghai, China
[e]State Key Laboratory of Synergistic Chem-Bio Synthesis, Shanghai Jiao Tong University, Shanghai, China

## Abstract

As widely used neuron model in Spiking Neural Networks (SNNs), the Leaky Integrate-and-Fire (LIF) model assumes the linear summation of injected currents. However, recent studies have revealed that a biological neuron can integrate inputs nonlinearly and perform computations such as XOR while an LIF neuron cannot. To bridge this gap, we propose the Dendritic LIF (DLIF) model, which incorporates a bilinear dendritic integration rule derived from neurophysiological experiments. At the single-neuron level, we theoretically demonstrate that a DLIF neuron can capture input correlations, enabling it to perform nonlinear classification tasks. At the network level, we prove that DLIF neurons can preserve and propagate correlation structures from the input layer to the readout layer. These theoretical findings are further confirmed by our numerical experiments. Extensive experiments across diverse architectures—including ResNet, VGG, and Transformer—demonstrate that DLIF achieves state-of-the-art performance on static (CIFAR-10/100, ImageNet) and neuromorphic (DVS-Gesture, DVS-CIFAR10) benchmarks, surpassing LIF and other advanced alternatives while maintaining comparable computational cost. This work provides a biologically plausible and computationally powerful spiking neuron model, paving the way for next-generation brain-inspired computing. All source code are available at https://github.com/majingyang0119/DLIF.

## 1 Introduction

Spiking neural networks (SNNs) are increasingly recognized as the next generation of neural network paradigm that closely emulates biological neural systems through discrete spike-based communication between neurons (Maass, 1997). Unlike traditional artificial neural networks (ANNs), which operate on continuous-valued activations (Deng et al., 2020), SNNs employ event-driven computation via discrete spikes. This fundamental difference enables SNNs to achieve significantly more efficient and sparse data processing, offering substantial energy efficiency advantages over conventional ANNs (Roy et al., 2019; Davies et al., 2018; Pei et al., 2019; Ma et al., 2022).

Most SNNs employ the Leaky Integrate-and-Fire (LIF) neuron model, a simplified abstraction of biological neurons that omits dendritic processing. In biological neurons, dendrites receive and integrate multiple input currents before transmitting them to the soma. The LIF model assumes the linear summation of the input currents. However, numerous studies have revealed that the integration process on the dendrites is nonlinear (Polsky et al., 2004; Poirazi et al., 2003; Ujfalussy et al., 2018; Beniaguev et al., 2021), which plays a critical role in complex computations of biological

---
[1]Corresponding author, songting@stju.edu.cn, or zdz@sjtu.edu.cn

neurons, such as direction selectivity (Branco et al., 2010), coincidence detection (Agmon-Snir et al., 1998), and logical operations (Gidon et al., 2020). Consequently, incorporating nonlinear dendritic integration features into spiking neuron models is an increasingly important direction in brain-inspired computing (Pagkalos et al., 2024; Acharya et al., 2022).

In this paper, we propose a novel spiking neuron model, termed the Dendritic Leaky Integrate-and-Fire (DLIF) model, which is based a bilinear dendritic integration rule observed in recent experiments (Hao et al., 2009; Li et al., 2014; 2019). Theoretically, we show that at the single-neuron level a DLIF neuron can capture input correlations, while at the network level DLIF neurons preserve and propagate correlation structures across layers, with these results further validated by numerical experiments. We also show that, across various tasks and deep neural network architectures, using DLIF models can significantly improve the performance of SNNs compared to those with LIF models and other spiking neuron models, with no significant increase in computational cost.

The main contributions of this paper are summarized as follows:

1. We propose the Dendritic Leaky Integrate-and-Fire (DLIF) model, a biologically plausible spiking neuron model that incorporates a bilinear dendritic integration rule observed in neurophysiological experiments.

2. We theoretically establish, and numerically confirm, that DLIF neurons can capture input correlations to perform nonlinear classification at the single-cell level, and preserve and propagate these correlations through the network.

3. We demonstrate that, across multiple architectures, DLIF neurons achieve an average accuracy of 85.18%, with a 1.23% improvement over conventional LIF-based SNNs (83.95%), and set state-of-the-art performance on both static and dynamic vision benchmarks. This improvement is obtained with only a 0.17 mJ energy overhead (a 3.05% relative increase).

4. We show that the DLIF models possess computational advantages comparable to those of other advanced spiking neuron models, including PLIF, GLIF, EIF, QIF and DH-LIF.

## 2 RELATED WORK

**Bilinear Neural Networks.** Several studies have explored bilinear neural networks in conventional ANNs. A line of work has focused on feature fusion and pooling for visual recognition (Lin et al., 2015; Gao et al., 2016; Kong & Fowlkes, 2017). Another direction has investigated bilinear neurons as architectural primitives: one-rank bilinear neurons (Yun et al., 2019), pixel-wise bilinear filters (Zoumpourlis et al., 2017), and bilinear networks with stabilized training strategies (Fan et al., 2025a). In addition, bilinear formulations have also been applied in other domains, including MRI reconstruction (Ahmed et al., 2022) and low-rank structures (Pearce et al., 2025). (Qi & Wang, 2022) further highlighted that bilinear networks can achieve substantially higher efficacy and efficiency than conventional neural networks. Our work differs by introducing bilinear dendritic integration into the spiking neural network framework and providing a theoretical analysis of its role in preserving input correlations, a perspective absent in prior bilinear ANN models.

**Models and Algorithms Inspired by Dendritic Computation.** Recent studies have increasingly incorporated dendritic computation principles into machine learning frameworks. Some works have drawn inspiration from dendritic cable theory, local learning rules, and dendritic event-based processing (Bicknell & Häusser, 2021; Payeur et al., 2021; Sacramento et al., 2018; Yang et al., 2021). Others have focused on network structures inspired by dendritic compartmentalization and connectivity (Guerguiev et al., 2017; Chavlis & Poirazi, 2025; Gao et al., 2018). Meanwhile, practical applications of dendritic integration have been demonstrated in diverse architectures, such as convolutional networks with dendritic modules (Liu et al., 2024) and dendritic artificial neural networks (Egrioglu & Bas, 2024). Together, these works highlight the growing importance of dendritic principles as a powerful source of inspiration for advancing machine learning.

**Neuron Models in SNNs.** Various extensions of the standard LIF model have been proposed to enhance the representational capacity of SNNs. Some works introduce additional flexibility in neuronal dynamics, such as learnable time constants or adaptive thresholds (Fang et al., 2021; Bellec et al., 2020; Feng et al., 2022; Chen et al., 2022). Others enrich the computational structure of

spiking neurons through gating mechanisms, soft reset strategies, or membrane potential rectifiers (Yao et al., 2022; Guo et al., 2022b). Multi-compartment and multi-branch models further capture dendritic or temporal heterogeneity (Zheng et al., 2024; Wang et al., 2025a; Liu et al., 2025b), while multi-synaptic formulations enable simultaneous integration at different scales (Fan et al., 2025b). These diverse extensions underline the central role of neuron model design in advancing the power of SNNs. However, to the best of our knowledge, no prior work has sought to optimize spiking neuron models through the bilinear form of dendritic integration.

## 3    DENDRITIC LEAKY INTEGRATE-AND-FIRE (DLIF) MODEL

Biological neurons have complex dendritic structures that are responsible for receiving multiple external inputs, integrating them, and transmitting the processed signals to the soma (Stuart et al., 2016). The spiking neuron model faithfully replicates key functional properties of biological neurons through temporal integration of input signals and subsequent generation of output spikes (Gerstner et al., 2014). The sub-threshold somatic membrane potential $V(t)$ of a spiking neuron is always governed by:

$$\mu \frac{dV(t)}{dt} = -(V(t) - V_{rest}) + RI(t),\tag{1}$$

where $\mu$ represents the time constant, $V_{rest}$ is the resting potential, $R$ denotes the resistance, and $I(t)$ is the input current. When $V(t)$ reaches a certain firing threshold $V_{th}$, the neuron emits a spike and resets the potential back to $V_{rest}$. The resulting output spike train $Y(t)$ is formally expressed as $Y(t) = \sum_i \delta(t - t^i)$, where $\delta$ is the Dirac delta function, and $t^i$ marks the $i$-th firing time of the neuron. As the commonly used model in SNNs, the LIF model assumes linear summation of external inputs (Burkitt, 2006), i.e.

$$I(t) = \sum_i \boldsymbol{w}_i \boldsymbol{s}_i(t) = \boldsymbol{w}^T \boldsymbol{s}(t).\tag{2}$$

where $\boldsymbol{w} = (\boldsymbol{w}_i)_{i=1}^n$ denotes the synaptic weight from the pre-synaptic neurons to the target post-synaptic neuron, and $\boldsymbol{s}(t) = (\boldsymbol{s}_i(t))_{i=1}^n$ represents the $\{0, 1\}$ spike trains from pre-synaptic neuron. However, biological experiments indicate that dendrites integrate inputs in a nonlinear manner (Polsky et al., 2004; Spruston, 2008). This dendritic nonlinearity is essential for various computational functions, such as direction selectivity (Branco et al., 2010), coincidence detection (Agmon-Snir et al., 1998), and logical operations (Gidon et al., 2020). Consequently, the linear dendritic integration mechanism of LIF models fails to fully capture the complex characteristics of biological neurons and cannot perform the rich nonlinear computations. To address this limitation, we propose a novel spiking neuron model, termed the Dendritic Leaky Integrate-and-Fire (DLIF) model.

### 3.1    FORMULATION OF THE DLIF MODEL

Recent neurophysiological experiments and theoretical analysis have demonstrated that the dendritic integration of synaptic inputs by a single neuron is not linear, but conforms to a bilinear form (Hao et al., 2009; Li et al., 2014; 2019). This bilinear integration property can be characterized by considering two synaptic inputs $a$ and $b$, where the dendritic integration yields not just the linear sum $a + b$ but includes an additional bilinear interaction term $kab$. Thus, the total integrated input becomes $a + b + kab$. Here, $k$ is referred to as the bilinear dendritic integration coefficient, which is independent of the input intensities and only dependent on the relative spatial positions of the two synaptic inputs. Consequently, when a neuron receives multiple synaptic input spike trains $\boldsymbol{s}(t) = (\boldsymbol{s}_i(t))_{i=1}^n$ with connection weights $\boldsymbol{w}$, there will be additional bilinear integration terms $\boldsymbol{s}_i(t)\boldsymbol{s}_j(t)$ ($1 \leq i < j \leq n$), associated with a symmetric bilinear coefficient matrix $\boldsymbol{K} = (\boldsymbol{K}_{ij})_{i,j=1}^n$ whose diagonal entries are zero. The integrated input can then be expressed as:

$$I(t) = \sum_{i=1}^n \boldsymbol{w}_i \boldsymbol{s}_i(t) + \sum_{i=1}^n \sum_{j>i}^n 2\boldsymbol{K}_{ij} \boldsymbol{s}_i(t)\boldsymbol{s}_j(t) = \boldsymbol{w}^T \boldsymbol{s}(t) + \boldsymbol{s}^T(t)\boldsymbol{K}\boldsymbol{s}(t).\tag{3}$$

Note that $\boldsymbol{s}_i(t)\boldsymbol{s}_j(t)$ can be directly realized through an AND operation, Eq. (3) won't introduce any additional multiplication operations. This preserves SNNs' computational efficiency, as their

spike-based communication naturally favors additive operations over multiplicative (Roy et al., 2019). Then the dynamics of the somatic membrane potential in the DLIF model can be described as:

$$\tau \frac{dV(t)}{dt} = -(V(t) - V_{rest}) + R[\boldsymbol{w}^T \boldsymbol{s}(t) + \boldsymbol{s}^T(t) \boldsymbol{K} \boldsymbol{s}(t)]. \tag{4}$$

## 3.2 Theoretical Analysis of DLIF's Computational Advantages

We first demonstrate the advantage of the DLIF neuron model from a theoretical perspective. We consider a binary classification problem where each input sample is represented as a binary matrix, where each column corresponds to the spike trains of $N$ input neurons at a given time step: $\boldsymbol{S} = [\boldsymbol{s}(1), \boldsymbol{s}(2), \cdots, \boldsymbol{s}(\tau)] \in \{0,1\}^{N \times \tau}$, where $N \in \mathbb{N}$ and $N \geq 2$, and $\tau$ is the total time steps. Two input classes with distributions $\mathcal{D}_1$ and $\mathcal{D}_2$ have identical mean firing rates but distinct pairwise correlations:

$$\frac{1}{\tau} \mathbb{E}_{\boldsymbol{S} \sim \mathcal{D}_1}[\boldsymbol{S} \boldsymbol{1}_\tau] = \frac{1}{\tau} \mathbb{E}_{\boldsymbol{S} \sim \mathcal{D}_2}[\boldsymbol{S} \boldsymbol{1}_\tau] = \boldsymbol{c}$$
$$\frac{1}{\tau} \mathbb{E}_{\boldsymbol{S} \sim \mathcal{D}_1}[\boldsymbol{S} \boldsymbol{S}^T] = \boldsymbol{C}_1 \neq \frac{1}{\tau} \mathbb{E}_{\boldsymbol{S} \sim \mathcal{D}_2}[\boldsymbol{S} \boldsymbol{S}^T] = \boldsymbol{C}_2 \tag{5}$$

where $\boldsymbol{1}_\tau$ is the all-ones column vector with length $\tau$. We claim that a single DLIF neuron can discriminate between two input classes by exhibiting distinct firing rates in response to them. (Gerstner & Kistler, 2002)) have demonstrated that for spiking neuron models with dynamics of the form given in Eq. (1), in the input regime where the current is sufficient to elicit spiking, the output firing rate is proportional to the average input current. Therefore, in this regime, a significant difference in firing rates is equivalent to a difference in input currents. Suppose the time-averaged input current to DLIF neuron is denoted as $I$, we can obtain the following theorem (See proof in Section A.1).

**Theorem 1.** *Let two input spike train distributions $\mathcal{D}_1$ and $\mathcal{D}_2$ be defined as in Eq. (5). Then there always exists a bilinear coefficient matrix $\boldsymbol{K}$ which can distinguish two corresponding input currents to the DLIF neuron, i.e.,*

$$\delta I = |\mathbb{E}[I|\mathcal{D}_1] - \mathbb{E}[I|\mathcal{D}_2]| > 0.$$

*Moreover, under the constraint $\|\boldsymbol{K}\|_F \leq 1$, the optimal choice of $\boldsymbol{K}$ that maximizes $\delta I$ is given by*

$$\boldsymbol{K}^* = \pm \frac{\boldsymbol{C}_1 - \boldsymbol{C}_2}{\|\boldsymbol{C}_1 - \boldsymbol{C}_2\|_F}.$$

This theorem shows that DLIF neurons can exploit input correlation structures for classification via the bilinear matrix $\boldsymbol{K}$. However, for multi-layer SNNs, it is generally reasonable to assume that correlated inputs appear at the input layer. In what follows, we further show that a two-layer SNN with DLIF neurons can preserve input correlations in the hidden layer.

Building upon the same input spike train distributions as in Eq. (5), without loss of generality, we assume that $\|\boldsymbol{C}_1 - \boldsymbol{C}_2\|_F = 1$. These spike trains are encoded by $M$ hidden neurons, and we denote the output spike trains of the hidden neurons as $\boldsymbol{Y} = [\boldsymbol{y}(1), \boldsymbol{y}(2), \cdots, \boldsymbol{y}(\tau)] \in \{0,1\}^{M \times \tau}$. For each class, we define the correlation matrix $\boldsymbol{P}_c$ of $\boldsymbol{Y}$ as $\boldsymbol{P}_c = \frac{1}{\tau} \mathbb{E}_{\boldsymbol{S} \sim \mathcal{D}_c}[\boldsymbol{Y} \boldsymbol{Y}^T]$ for $c \in \{1, 2\}$, and denote the resulting matrices for LIF and DLIF neurons by $\boldsymbol{P}_1^{\text{LIF}}, \boldsymbol{P}_2^{\text{LIF}}, \boldsymbol{P}_1^{\text{DLIF}}, \boldsymbol{P}_2^{\text{DLIF}}$, respectively. Let $\boldsymbol{W} = [\boldsymbol{w}_1, \ldots, \boldsymbol{w}_M] \in \mathbb{R}^{M \times N}$ be the weight matrix and $\mathbf{K} = [\boldsymbol{K}_1, \ldots, \boldsymbol{K}_M]$ with $\boldsymbol{K}_m \in \mathbb{R}^{N \times N}$ be the bilinear coefficient matrices. We suppose that $\|\boldsymbol{w}_i\|_F \leq 1$ and $\|\boldsymbol{K}_i\|_F \leq 1$ for $i = 1, \cdots, M$. Then we obtain the following theorem (See proof in Section A.1).

**Theorem 2.** *Let the input spike trains be drawn from distributions $\mathcal{D}_1$ and $\mathcal{D}_2$ defined in Eq. (5). Then for any choice of synaptic weight matrices $\boldsymbol{W}^{\text{LIF}}$ and $\boldsymbol{W}^{\text{DLIF}}$, there exists bilinear coefficient matrices $\mathbf{K}$ for the DLIF neurons such that*

$$\|\boldsymbol{P}_1^{\text{DLIF}} - \boldsymbol{P}_2^{\text{DLIF}}\|_F \geq C > 0,$$

*and furthermore,*

$$\|\boldsymbol{P}_1^{\text{DLIF}} - \boldsymbol{P}_2^{\text{DLIF}}\|_F > \|\boldsymbol{P}_1^{\text{LIF}} - \boldsymbol{P}_2^{\text{LIF}}\|_F.$$

This theorem demonstrates that DLIF networks are more capable than LIF networks at preserving the correlation structures inherent in the input data. Combined with Theorem 1, which proves that the DLIF neuron model can classify inputs with correlated structures, these theoretical derivations collectively guarantee the superior computational and representational power of DLIF neurons over LIF neurons. Next, we further validate the above theorems through several numerical experiments.

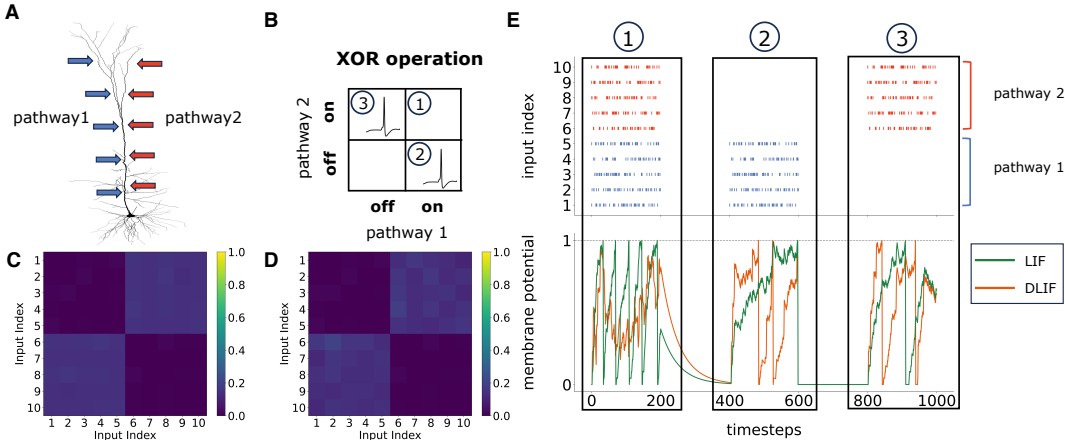

Figure 1: **A single DLIF model can effectively solve the XOR problem.**
(A): A neuron receives inputs from two distinct pathways (represented in blue and red), each containing five synaptic inputs.
(B): The XOR operation schematic: the neuron remains inactive when both pathways are on or off, but it fires when only one pathway is on, thereby implementing the XOR operation. It is trivial that the neuron is inactive when both pathways are off. Therefore, we only consider the other three input patterns, which are labeled as ①, ②, and ③, respectively.
(C): Expectation of the normalized difference in correlation matrices between two classes $\frac{C_1-C_2}{\|C_1-C_2\|_F}$.
(D): Expectation of the bilinear coefficient $K$ for the DLIF neuron.
(E): Three different patterns in the XOR problem. Pathway 1 includes synapse indices 1-5, while pathway 2 includes synapse indices 6-10. The DLIF model keeps silent when both pathways are activated (labeled as ①) and fires when receiving input patterns with only one pathway is activated (labeled as ② and ③), thus achieving the XOR operation (orange). In contrast, the LIF model still fires when both pathways are active (labeled as ①), thus it is unable to perform the XOR operation.

## 3.3 NUMERICAL VERIFICATION

**Verification of Theorem 1** We first verify Theorem 1 by showing that a single DLIF neuron can implement the XOR operation, consistent with recent biological findings that individual neurons can perform such computations (Gidon et al., 2020), whereas a standard LIF neuron cannot (Mostafa, 2017). In our simulation, the neuron receives inputs from two pathways (five synapses each) with identical Poisson firing rates (Fig. 1A). The target is to fire when exactly one pathway is active but remain silent otherwise (Fig. 1B). We group the three non-trivial input patterns into two classes: Class 1 (both pathways active) and Class 2 (only one pathway active). Under this setup, the spike trains of the two input classes follow a similar distribution as in Eq. (5). We train $w$ for LIF and both $w$ and $K$ for DLIF to minimize the mean squared error between actual and target firing rates. As shown in Fig. 1E, the DLIF model successfully performs XOR while the LIF model fails, and the learned bilinear matrices $K$ closely match $\frac{C_1-C_2}{\|C_1-C_2\|_F}$ (Fig. 1C, D), consistent with theoretical predictions.

**Verification of Theorem 2** To validate Theorem 2, we design controlled numerical experiments at both low and moderate input dimensionalities. First, a two-dimensional case provides a simple and interpretable setting, where two Poisson input spike trains are received by two hidden neurons which employs either LIF or DLIF neurons. The network is trained to maximize output correlation (details in the Appendix), and we compute the normalized correlation between output spike trains. As shown in Fig. 2A, DLIF neurons preserve input correlations substantially more effectively than LIF neurons. To further examine whether this advantage persists in more complex scenarios, we consider a ten-dimensional case where two distinct input classes with distributions $\mathcal{D}_1$ and $\mathcal{D}_2$ as in Eq. (5). These inputs are fed into a ten-dimensional hidden layer, and the training procedure is similar to the 2D case. As illustrated in Fig. 2B, DLIF neurons again yield significantly larger separation between the output correlation matrices of the two classes compared to LIF neurons. Together, these

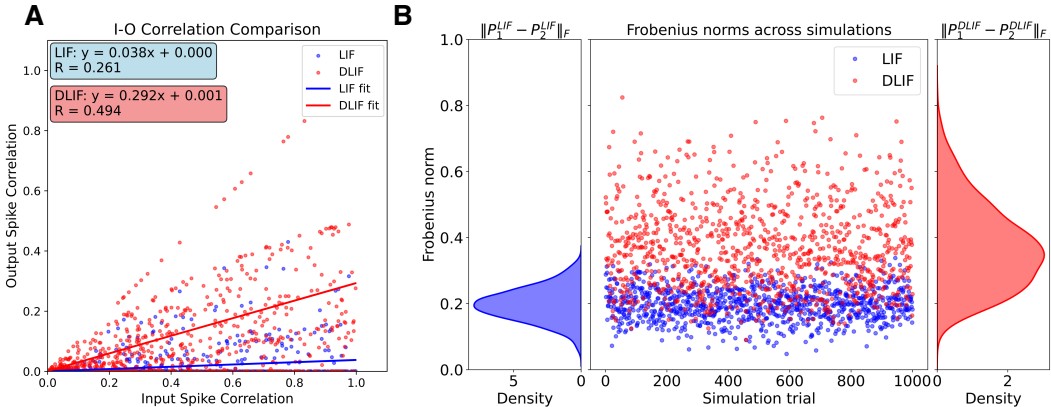

Figure 2: **DLIF neurons can preserve and propagate input correlations.**
(A): For the two-dimensional case, we compare the input-output correlation across 1000 simulation trials for both LIF and DLIF neurons. Each dot represents an individual trial. The regression slopes indicate that DLIF neurons preserve input correlations more effectively (slope = 0.292, R= 0.494) than LIF neurons (slope = 0.038, R=0.261).
(B): For the ten-dimensional case, we compare the Frobenius norm of the difference between the output correlation matrices of two input classes across 1,000 simulation trials for both LIF and DLIF neurons. Each dot corresponds to an individual trial. The scatter plot, together with the marginal distributions, demonstrates that DLIF neurons consistently produce a larger separation between input classes than LIF neurons.

results confirm the theoretical prediction in Theorem 2 that DLIF neurons more effectively amplify and propagate correlation differences across layers.

## 4 PERFORMANCE OF DLIF MODELS IN SNNS

We further investigate whether the advantage of DLIF models can be scaled to large-scale SNN architectures. The detailed SNN architectures and experimental setup used in this work are presented in the Appendix.

As shown in Eq. (3), the DLIF model introduces additional bilinear parameters $K$. Through both biological recordings and computational modeling, Li et al. (2019) reported that dendritic bilinear interactions are inherently sparse ($\approx 90\%$). Motivated by this, we adopt a sparsification scheme in which only a small fraction of coefficients are trainable. Specifically, we set sparsity level to 90%, a choice that is both biologically inspired and empirically validated by an ablation study as shown in Section 4.4. In the following experiments, we compare not only model accuracy but also parameter count, FLOPs, and energy consumption (detailed calculation methods are provided in the Appendix).

### 4.1 EXPERIMENTS IN LARGE-SCALE SNNS FOR STATIC DATASETS

We first evaluate the performance of the DLIF models in the SNNs on static image classification benchmarks, including CIFAR10 (Krizhevsky et al., 2009), CIFAR100 (Krizhevsky et al., 2009) and ImageNet (Deng et al., 2009) datasets. Additionally, we test the DLIF models' applicability across diverse network architectures and training methods. Specifically, we explore the integration of DLIF models in large-scale SNNs, including VGG (Simonyan & Zisserman, 2014), ResNet (He et al., 2016) and Transformer (Ashish et al., 2017). We also experiment using different learning paradigms including SLTT (Meng et al., 2023), ESG (Guo et al., 2022a), OTTT (Xiao et al., 2022), STBP-tdBN (Zheng et al., 2021), TET (Deng et al., 2022), ESL (Liu et al., 2025a), TSER (Yu et al., 2025a), FSTA (Yu et al., 2025b), SSSA (Wang et al., 2025b), Spike-driven Transformer (Yao et al., 2023a) and Meta-SpikeFormer (Yao et al., 2024), as proposed in previous works.

Table 1: Results on Static Datasets

| Dataset | Method | Network | Neuron | Params (M) | FLOPs (G) | Energy Cost (mJ) | Mean± Std (%) |
|---|---|---|---|---|---|---|---|
| CIFAR10 | SLTT | ResNet-18 | LIF | 12.08 | 1.82 | 1.77 | 94.44±0.21 |
| | | | DLIF | 12.20 | 1.83 | 1.78 | **95.51±0.47** |
| | STBP-tdBN | ResNet-19 | LIF | 12.61 | 1.94 | 1.88 | 93.16 |
| | | | DLIF | 12.78 | 1.95 | 1.89 | **94.35±0.27** |
| | ESL | ResNet-18 | LIF | 12.08 | 1.83 | 1.77 | 96.39 |
| | | | DLIF | 12.20 | 1.83 | 1.78 | **96.72±0.11** |
| | TSER | VGG-16 | LIF | 138.08 | 15.48 | 15.05 | 95.01±0.10 |
| | | | DLIF | 136.02 | 15.94 | 15.57 | **95.97±0.12** |
| | FSTA | ResNet-19 | LIF | 12.61 | 1.94 | 1.88 | 96.52±0.09 |
| | | | DLIF | 12.78 | 1.95 | 1.89 | **96.91±0.08** |
| | SSSA | Transformer | LIF | 5.57 | 1.21 | 1.04 | 96.10 |
| | | | DLIF | 5.68 | 1.28 | 1.10 | **96.81±0.17** |
| | Spike-driven Transformer | Transformer | LIF | 9.32 | 1.08 | 1.05 | 95.60 |
| | | | DLIF | 10.05 | 1.10 | 1.07 | **96.22±0.17** |
| CIFAR100 | SLTT | ResNet-18 | LIF | 12.17 | 1.82 | 1.77 | 74.38±0.30 |
| | | | DLIF | 13.21 | 1.83 | 1.78 | **76.89±0.29** |
| | ESG | VGG-16 | LIF | 138.44 | 15.50 | 15.07 | 70.18±0.09 |
| | | | DLIF | 146.86 | 16.04 | 15.60 | **73.52±0.16** |
| | OTTT | VGG-11 | LIF | 123.60 | 7.63 | 7.42 | 71.05±0.04 |
| | | | DLIF | 136.71 | 8.13 | 7.91 | **74.71±0.19** |
| | TSER | VGG-16 | LIF | 138.44 | 15.50 | 15.07 | 77.06±0.04 |
| | | | DLIF | 146.86 | 16.04 | 15.60 | **78.37±0.14** |
| | FSTA | ResNet-19 | LIF | 12.74 | 1.94 | 1.88 | 80.42±0.09 |
| | | | DLIF | 12.91 | 1.95 | 1.89 | **80.97±0.12** |
| | SSSA | Transformer | LIF | 5.57 | 1.21 | 1.04 | 80.10 |
| | | | DLIF | 5.63 | 1.30 | 1.12 | **80.58±0.09** |
| | Spike-driven Transformer | Transformer | LIF | 9.35 | 1.08 | 1.05 | 78.40 |
| | | | DLIF | 10.12 | 1.10 | 1.07 | **79.46±0.32** |
| ImageNet | TET | ResNet-34 | LIF | 21.80 | 3.66 | 3.56 | 64.79 |
| | | | DLIF | 23.42 | 3.85 | 3.74 | **67.32±0.39** |
| | STBP-tdBN | ResNet-34 | LIF | 21.80 | 3.66 | 3.56 | 63.72 |
| | | | DLIF | 23.42 | 3.85 | 3.74 | **66.79±0.53** |
| | ESL | VGG-16 | LIF | 138.44 | 15.50 | 15.00 | 74.32 |
| | | | DLIF | 146.86 | 16.04 | 15.60 | **75.11±0.28** |
| | TSER | ResNet-34 | LIF | 21.80 | 3.66 | 3.56 | 73.16±0.15 |
| | | | DLIF | 23.42 | 3.85 | 3.74 | **73.82±0.29** |
| | FSTA | ResNet-34 | LIF | 21.80 | 3.66 | 3.56 | 70.23±0.12 |
| | | | DLIF | 23.42 | 3.85 | 3.74 | **71.06±0.15** |
| | SSSA | Transformer | LIF | 53.7 | 36.75 | 35.75 | 80.23 |
| | | | DLIF | 57.37 | 37.04 | 36.11 | **80.75±0.24** |
| | Meta-SpikeFormer | Transformer | LIF | 55.40 | 53.92 | 52.40 | 80.00 |
| | | | DLIF | 58.73 | 54.35 | 53.68 | **80.57±0.28** |

Bold values represent the best results for each method

As summarized in Table 1, our results reveal consistent and substantial accuracy improvements attributable to DLIF models across all configurations. Specifically, DLIF-based SNNs achieve absolute accuracy gains of 0.33%–1.19% on CIFAR-10, with even more pronounced improvements of 0.48%–3.66% on CIFAR-100 and 0.52%–3.07% on ImageNet. Notably, these performance enhancements come with only minimal energy overhead, quantified as just 0.23 mJ average increase (2.61% relative to LIF implementations).

Table 2: Results on Neuromorphic Datasets

| Dataset | Method | Network | Neuron | Params (M) | FLOPs (G) | Energy Cost (mJ) | Mean± Std (%) |
|---|---|---|---|---|---|---|---|
| DVS-Gesture | SLTT | VGG-11 | LIF | 123.24 | 7.61 | 7.40 | 98.50±0.21 |
| | | | DLIF | 125.96 | 7.85 | 7.63 | **98.92±0.24** |
| | OTTT | VGG-11 | LIF | 123.24 | 7.61 | 7.40 | 96.88 |
| | | | DLIF | 125.96 | 7.85 | 7.63 | **97.43±0.46** |
| | STBP-tdBN | ResNet-17 | LIF | 11.74 | 1.71 | 1.67 | 96.87 |
| | | | DLIF | 11.87 | 1.72 | 1.68 | **98.05±0.41** |
| | SSNN | VGG-9 | LIF | 27.48 | 2.13 | 2.08 | 94.91 |
| | | | DLIF | 28.55 | 2.23 | 2.18 | **96.27±0.32** |
| | Spike-driven Transformer | Transformer | LIF | 2.59 | 0.36 | 0.35 | 99.30 |
| | | | DLIF | 3.02 | 0.37 | 0.36 | **99.43±0.27** |
| DVS-CIFAR10 | SLTT | VGG-11 | LIF | 123.24 | 7.61 | 7.40 | 82.20±0.95 |
| | | | DLIF | 125.96 | 7.85 | 7.63 | **83.74±0.62** |
| | STBP-tdBN | ResNet-19 | LIF | 12.61 | 1.94 | 1.88 | 67.8 |
| | | | DLIF | 12.78 | 1.95 | 1.89 | **70.88±0.45** |
| | SSNN | VGG-9 | LIF | 27.48 | 2.13 | 2.08 | 78.57 |
| | | | DLIF | 28.55 | 2.23 | 2.18 | **80.85±0.42** |
| | FSTA | ResNet-20 | LIF | 13.57 | 2.21 | 2.17 | 82.70±0.10 |
| | | | DLIF | 13.85 | 2.38 | 2.31 | **82.98±0.13** |
| | SSSA | Transformer | LIF | 1.52 | 0.54 | 0.52 | 82.30 |
| | | | DLIF | 1.84 | 0.56 | 0.53 | **82.91±0.15** |
| | Spike-driven Transformer | Transformer | LIF | 2.59 | 0.36 | 0.35 | 80.00 |
| | | | DLIF | 3.02 | 0.37 | 0.36 | **81.76±0.27** |

Bold values represent the best results for each method

## 4.2 Experiments in Large-scale SNNs for Neuromorphic Datasets

In contrast to static image datasets, neuromorphic datasets like DVS-Gesture (Amir et al., 2017) and DVS-CIFAR10 (Li et al., 2017) naturally encode temporal information, thereby better show-casing SNNs' inherent advantages in processing spatiotemporal patterns. We further evaluate the performance of SNNs using DLIF models versus LIF models across different network architectures including including VGG , ResNet and Transformer, and different learning paradigms such as SLTT, OTTT, STBP-tdBN, SSNN, Spike-driven Transformer and FSTA.

The experimental results in Table 2 demonstrate consistent performance gains when using DLIF models. Specifically, on DVS-Gesture, DLIF-based SNNs achieve 0.13%-1.36% higher accuracy, while on the more complex DVS-CIFAR10, the improvements reach 0.28%-3.08%. Importantly, these significant accuracy gains come with only 0.10 mJ average energy increase (3.24% relative to LIF models), further validating DLIF's practical utility in neuromorphic computing applications.

In addition, we further compare the training time and memory cost of DLIF and LIF models in Section A.3. The results show that DLIF increases per-epoch training time and GPU memory usage by roughly 10%, but this overhead remains acceptable for large-scale SNN training.

Beyond static and neuromorphic benchmarks, we further evaluate DLIF models in reinforcement learning (RL) by integrating them into a deep spiking Q-network (DSQN) (Chen et al., 2024). On Atari games, the DLIF-based DSQN outperforms its LIF counterpart, demonstrating the flexibility of DLIF models to adapt effectively across diverse task paradigms (see Section A.4 for details).

## 4.3 Comparisons with Other Spiking Neuron Models

Several studies have proposed modifications to the existing LIF models in SNNs. For instance, Fang et al. (2021) introduced the Parametric Leaky Integrate-and-Fire (PLIF) model, which included learnable time constants to enhance heterogeneity. Yao et al. (2022) proposed the Gated Leaky Integrate-and-Fire (GLIF) model, incorporating gating units into LIF models to improve their rep-

Table 3: Comparisons with Other Point Spiking Neuron Models.

| Neuron Model | Accuracy CIFAR-10(%) | Accuracy CIFAR-100(%) | Accuracy ImageNet(%) | Accuracy DVS-CIFAR10(%) | Accuracy DVS-Gesture(%) |
|---|---|---|---|---|---|
| PLIF | 93.50 | - | 69.26 | 74.80 | 97.92 |
| GLIF | 95.03±0.08 | 77.35±0.07 | 69.09 | 78.10 | - |
| QIF | 92.98±0.14 | 75.91±0.08 | 67.49±0.25 | 73.27±0.19 | 96.18±0.11 |
| EIF | 93.08±0.17 | 76.18±0.15 | 67.14±0.30 | 76.27±0.39 | 97.01±0.18 |
| DLIF | **95.78±0.21** | **78.27±0.39** | **71.27±0.24** | **80.46±0.17** | **98.61±0.31** |

Bold values represent the best results for each dataset;− indicates result is not reported

Table 4: Comparisons with DH-LIF.

| Neuron Model | Accuracy on SHD(%) | Accuracy on SSC(%) |
|---|---|---|
| DH-LIF | 92.10 | 82.46 |
| DLIF | **92.71** | **83.13** |

Bold values represent the best results for each dataset

resentation capacity. In addition to the PLIF and GLIF models, other variants introduce nonlinear operations to the LIF model, such as the Quadratic Integrate-and-Fire (QIF) model and the Exponential Integrate-and-Fire (EIF) model (Gerstner et al., 2014). The detailed of the dynamics of the QIF and EILF model are shown in the Appendix. To ensure fair comparison, we adopt the same network architectures and hyperparameter configurations as in prior works. Across CIFAR-10, CIFAR-100, ImageNet, DVS-CIFAR10, and DVS-Gesture, our results (Table 3) show that DLIF consistently outperforms these advanced point-neuron models.

In addition to point-neuron models, we also compare DLIF with a multi-compartment spiking neuron model—the DH-LIF proposed by (Zheng et al., 2024), which incorporates temporal dendritic heterogeneity. We follow the experimental setup in (Zheng et al., 2024) and conduct comparisons on two spiking speech recognition datasets SHD and SSC (Cramer et al., 2020). Under comparable parameter settings, DLIF consistently surpasses DH-LIF on both tasks, demonstrating its effectiveness relative to dendritic neuron models as well (Table 4).

## 4.4 ABLATION STUDY

**Sparsity Level** We conducted a systematic ablation study by varying sparsity levels from 0% to 100% on the CIFAR-100 dataset with the ResNet-18 architecture trained using SLTT. As summarized in Table 5, the ACC/FLOPs ratio consistently reaches its maximum at 90%. This indicates that 90% sparsity provides the most favorable trade-off between computational efficiency and predictive performance. Combined with biological evidence that dendritic bilinear interactions are naturally sparse at about 90% (Li et al., 2019), these results justify our adoption of 90% sparsity.

Table 5: Ablation Study of Sparsity Level

| Sparsity level(%) | 0 | 15 | 30 | 45 | 60 |
|---|---|---|---|---|---|
| FLOPs(G) | 1.92 | 1.905 | 1.89 | 1.875 | 1.86 |
| ACC(%) | 78.67 | 78.45 | 78.14 | 77.42 | 77.26 |
| ACC/FLOPs((%/G) | 40.97 | 41.18 | 41.34 | 41.29 | 41.54 |
| Sparsity level(%) | 75 | 85 | 90 | 95 | 100 |
| FLOPs(G) | 1.845 | 1.835 | 1.83 | 1.825 | 1.82 |
| ACC(%) | 76.33 | 76.38 | 76.89 | 74.61 | 74.38 |
| ACC/FLOPs((%/G) | 41.67 | 41.62 | **42.02** | 40.88 | 40.87 |

Bold values represent the best results

**Bilinear Coefficients** To further illustrate the role of the bilinear coefficients $K$ in the DLIF model, we conduct ablation studies to assess their impact. As shown in Section A.5, removing the bilinear coefficients—either before or after training—consistently reduces test accuracy, confirming their critical importance in the DLIF model. In addition, our structured- and low-rank-parameterization experiments show that low-rank formulations, while offering stronger compression, lead to clearly weaker performance, whereas the diagonal-block and random sparse formulations perform comparably under matched sparsity levels. Together with sparsity-level ablations showing that performance peaks around 90% sparsity, these findings reinforce the central role of the bilinear coefficients $K$ in the DLIF model and highlight that a 90% random sparse matrix provides a simple yet effective parameterization.

## 5    DISCUSSION AND CONCLUSION

This paper proposed the DLIF model, which incorporates a biologically inspired dendritic bilinear integration rule into spiking neurons. While (Li et al., 2019) investigated bilinear dendritic integration using a conductance-based model with voltage-dependent synaptic dynamics, this formulation is difficult to scale to large SNNs. In contrast, our DLIF model adopts a current-based abstraction that removes these biophysical dependencies and enables efficient, scalable implementation while preserving the bilinear rule. We further provided theoretical guarantees and numerical verification that DLIF neurons can exploit input correlations for nonlinear computation and preserve correlation structures across layers. Experiments on static, neuromorphic and RL benchmarks consistently showed that DLIF can achieve superior performance over LIF and other advanced spiking models, with minimal additional energy cost.

There remain some important avenues for further research. One direction is to extend DLIF beyond vision and RL tasks to natural language processing, especially in light of the rapid progress of large language models. Another direction concerns hardware adaptation. While DLIF models demonstrate strong algorithmic efficiency, adopting them onto neuromorphic hardware will be crucial to fully exploit the low-power and low-latency advantages of SNNs. Addressing these open challenges would further enhance the applicability and impact of DLIF models.

In summary, we propose a novel spiking neuron model for SNNs that enhances computational capabilities at both the single-neuron and network levels. We believe this work provides a solid foundation for the design and application of future brain-inspired computing.

## 6    REPRODUCIBILITY STATEMENT

We ensure reproducibility at several levels. First, the DLIF model is clearly described in the main text, including its mathematical formulation and theoretical analysis. Second, the assumptions and complete proofs of all theorems are provided in Appendix. Third, experimental settings—including datasets, architectures, hyperparameters, and training paradigms—are specified in Appendix.

## 7    THE USE OF LARGE LANGUAGE MODELS (LLMS)

We used LLMs to polish the manuscript, for example by improving phrasing and checking spelling and grammar. LLMs were also employed to assist in literature search and discovery, such as by providing keywords to retrieve related works. However, the core ideas, methodology, and contributions of this paper were conceived independently and did not rely on LLMs.

## 8    ACKNOWLEDGMENT

This work was supported by STI2030-Major Projects 2021ZD0200204 (D.Z., S.L.); Shanghai Municipal Commission of Science and Technology with Grant No. 24JS2810400 (D.Z., S.L.); National Natural Science Foundation of China Grant 12271361, 12250710674 (S.L.); National Natural Science Foundation of China with Grant No. 92570202, 12225109 (D.Z.); Lingang Laboratory Grant No. LG-QS202202-01 (S.L., D.Z.); and the Student Innovation Center at Shanghai Jiao Tong University (J.M., C.L., S.L., D.Z.).

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

# A  APPENDIX

## A.1  PROOFS OF THEOREMS

We consider a binary classification problem where each input sample is a binary matrix: $\boldsymbol{S} = [\boldsymbol{s}(1), \boldsymbol{s}(2), \cdots, \boldsymbol{s}(\tau)] \in \{0, 1\}^{N \times \tau}$, where $N \in \mathbb{N}$ and $N \geq 2$, and $\tau$ is the total number of discrete time steps. Two input classes with distributions $\mathcal{D}_1$ and $\mathcal{D}_2$ have identical mean firing rates but distinct pairwise correlations:

$$
\begin{aligned}
\frac{1}{\tau}\mathbb{E}_{\boldsymbol{S} \sim \mathcal{D}_1}[\boldsymbol{S}\mathbf{1}_\tau] &= \frac{1}{\tau}\mathbb{E}_{\boldsymbol{S} \sim \mathcal{D}_2}[\boldsymbol{S}\mathbf{1}_\tau] = \boldsymbol{c} \\
\frac{1}{\tau}\mathbb{E}_{\boldsymbol{S} \sim \mathcal{D}_1}[\boldsymbol{S}\boldsymbol{S}^T] &= \boldsymbol{C}_1 \neq \frac{1}{\tau}\mathbb{E}_{\boldsymbol{S} \sim \mathcal{D}_2}[\boldsymbol{S}\boldsymbol{S}^T] = \boldsymbol{C}_2
\end{aligned}
\tag{A1}
$$

where $\mathbf{1}_\tau$ is the all-ones column vector with length $\tau$. (Gerstner & Kistler, 2002) have demonstrated that for spiking neuron models with dynamics of the form given in $\tau\frac{dV(t)}{dt} = -(V(t)-V_{rest})+RI(t)$, in the input regime where the current is sufficient to elicit spiking, the firing rate is proportional to the average input current. Assuming that the average firing rate is sufficiently high to induce spiking, we obtain the following theorem

**Theorem A1.** *Let two input spike train distributions $\mathcal{D}_1$ and $\mathcal{D}_2$ be defined as in Eq. (5). Then there always exists a bilinear coefficient matrix $\boldsymbol{K}$ which can distinguish two corresponding input currents to the DLIF neuron, i.e.,*

$$
\delta I = |\mathbb{E}[I|\mathcal{D}_1] - \mathbb{E}[I|\mathcal{D}_2]| > 0.
$$

*Moreover, under the constraint $\|\boldsymbol{K}\|_F \leq 1$, the optimal choice of $\boldsymbol{K}$ that maximizes $\delta I$ is given by*

$$
\boldsymbol{K}^* = \pm\frac{\boldsymbol{C}_1 - \boldsymbol{C}_2}{\|\boldsymbol{C}_1 - \boldsymbol{C}_2\|_F}.
$$

*Proof.* Let $\boldsymbol{w}$ and $\boldsymbol{K}$ denote the weight vector and the bilinear coefficient matrix of a DLIF neuron, respectively. With a certain input $\boldsymbol{S}$, the total integrated input current is defined as in Eq. (3). Then the average input current is:

$$
I = \frac{1}{\tau}\sum_{t=1}^{\tau} I(t) = \frac{1}{\tau}(\boldsymbol{w}^T\boldsymbol{S}\mathbf{1}_\tau + tr(\boldsymbol{S}^T\boldsymbol{K}\boldsymbol{S}))
\tag{A2}
$$

Then the difference of the input current between the two input classes is:

$$
\delta I = \big|\mathbb{E}[I|\mathcal{D}_1] - \mathbb{E}[I|\mathcal{D}_2]\big| = \big|tr(\boldsymbol{K}(\boldsymbol{C}_1 - \boldsymbol{C}_2))\big|
\tag{A3}
$$

Since $\boldsymbol{C}_1 \neq \boldsymbol{C}_2$, there exists at two entries $k = (\boldsymbol{C}_1 - \boldsymbol{C}_2)_{ij} = (\boldsymbol{C}_1 - \boldsymbol{C}_2)_{ji} \neq 0$ ($\boldsymbol{C}_1 - \boldsymbol{C}_2$ is symmetric). Consider a bilinear coefficient matrix $\boldsymbol{K}$ that places nonzero values only on these two entry (e.g., $\boldsymbol{K}_{ij} = \boldsymbol{K}_{ji} = 1$ and all other entries zero). Then $\delta I = 2k > 0$, which establishes the existence of $\boldsymbol{K}$ such that $\delta I > 0$.

Furthermore, under the normalization constraint $\|\boldsymbol{K}\|_F = 1$, we have

$$
\big|tr(\boldsymbol{K}(\boldsymbol{C}_1 - \boldsymbol{C}_2))\big| \leq \|\boldsymbol{K}\|_F \|\boldsymbol{C}_1 - \boldsymbol{C}_2\|_F = \|\boldsymbol{C}_1 - \boldsymbol{C}_2\|_F.
\tag{A4}
$$

Equality holds if and only if

$$
\boldsymbol{K} = \pm\frac{\boldsymbol{C}_1 - \boldsymbol{C}_2}{\|\boldsymbol{C}_1 - \boldsymbol{C}_2\|_F},
$$

in which case $\delta I$ attains its maximal value $\|\boldsymbol{C}_1 - \boldsymbol{C}_2\|_F$. Therefore, not only is separation guaranteed, but there also exists an optimal $\boldsymbol{K}$ that maximizes the expected difference in input current, ensuring the clearest distinction between the two classes. $\square$

At the network level, suppose the two input classes are still characterized by Eq. (A1). These inputs are encoded by $M$ hidden neurons, and we denote the input current to the hidden neurons as $\boldsymbol{I} = [\boldsymbol{i}(1), \boldsymbol{i}(2), \cdots, \boldsymbol{i}(\tau)] \in \mathbb{R}^{M \times \tau}$. Assume that the two classes exhibit significantly different correlation structures; without loss of generality, we let $\|\boldsymbol{C}_1 - \boldsymbol{C}_2\|_F = 1$. For each class, we define the correlation matrix of $\boldsymbol{Q}_c$ of $\boldsymbol{I}$ as $\boldsymbol{Q}_c = \frac{1}{\tau}\mathbb{E}_{\boldsymbol{S} \sim \mathcal{D}_c}[\boldsymbol{I}\boldsymbol{I}^T]$ for $c \in \{1, 2\}$, and denote the

resulting matrices for LIF and DLIF neurons by $\boldsymbol{Q}_1^{\text{LIF}}, \boldsymbol{Q}_2^{\text{LIF}}, \boldsymbol{Q}_1^{\text{DLIF}}, \boldsymbol{Q}_2^{\text{DLIF}}$, respectively. Let $\boldsymbol{W} = [\boldsymbol{w}_1, \ldots, \boldsymbol{w}_M] \in \mathbb{R}^{M \times N}$ be the weight matrix and $\mathbf{K} = [\boldsymbol{K}_1, \ldots, \boldsymbol{K}_M]$ with $\boldsymbol{K}_m \in \mathbb{R}^{N \times N}$ be the bilinear coefficient matrices. Without loss of generality, assume that the LIF and DLIF neurons share the same synaptic weight matrix $\boldsymbol{W} = [\boldsymbol{w}_1, \ldots, \boldsymbol{w}_M] \in \mathbb{R}^{M \times N}$. Suppose further that the weight vectors and bilinear coefficient matrices are normalized such that $\|\boldsymbol{w}_i\|_F \leq 1$ and $\|\boldsymbol{K}_i\|_F \leq 1$ for all $i = 1, \ldots, M$. Then we obtain the following lemma:

**Lemma A1.** *Let the input spike trains be drawn from distributions $\mathcal{D}_1$ and $\mathcal{D}_2$ defined in Eq. (5). Then for any choice of synaptic weight matrices $\boldsymbol{W}$, there exists bilinear coefficient matrices $\mathbf{K}$ for the DLIF neurons such that*

$$\|\boldsymbol{Q}_1^{\text{DLIF}} - \boldsymbol{Q}_2^{\text{DLIF}}\|_F \geq C > 0.$$

*and*

$$\|\boldsymbol{Q}_1^{\text{DLIF}} - \boldsymbol{Q}_2^{\text{DLIF}}\|_F > \|\boldsymbol{Q}_1^{\text{LIF}} - \boldsymbol{Q}_2^{\text{LIF}}\|_F.$$

*Proof.* The input current to $i$th hidden neurons are defined as

$$\boldsymbol{I}_{i,:}^{\text{LIF}} = \boldsymbol{w}_i^T \boldsymbol{S}, \qquad \boldsymbol{I}_{i,:}^{\text{DLIF}} = \boldsymbol{w}_i^T \boldsymbol{S} + \Phi_i(\boldsymbol{S}), \tag{A5}$$

where $\Phi_i(\boldsymbol{S}) = [\boldsymbol{s}(1)^\top \boldsymbol{K}_i \boldsymbol{s}(1), \ldots, \boldsymbol{s}(\tau)^\top \boldsymbol{K}_i \boldsymbol{s}(\tau)]$. For class $c \in \{1, 2\}$, define the correlation matrix as

$$(\boldsymbol{Q}_c^*)_{ij} = \mathbb{E}_{\boldsymbol{S} \sim \mathcal{D}_c}[\boldsymbol{I}_{i,:}^*, (\boldsymbol{I}_{j,:}^*)^T], \tag{A6}$$

where $* \in \{\text{LIF}, \text{DLIF}\}$.

**Step 1 (LIF case)**

For LIF neurons we have

$$(\boldsymbol{Q}_c^{\text{LIF}})_{ij} = \mathbb{E}_{\boldsymbol{S} \sim \mathcal{D}_c}[\boldsymbol{I}_{i,:}^{\text{LIF}}(\boldsymbol{I}_{j,:}^{\text{LIF}})^T] = \mathbb{E}_{\boldsymbol{S} \sim \mathcal{D}_c}[\boldsymbol{w}_i^T \boldsymbol{S} \boldsymbol{S}^T \boldsymbol{w}_j] = \boldsymbol{w}_i^T \boldsymbol{C}_c \boldsymbol{w}_j \tag{A7}$$

where $\boldsymbol{C}_c = \mathbb{E}_{\boldsymbol{S} \sim \mathcal{D}_c}[SS^T]$ is the input correlation matrix of class $c$. Hence

$$\Delta_{ij}^{\text{LIF}} = (\boldsymbol{Q}_1^{\text{LIF}} - \boldsymbol{Q}_2^{\text{LIF}})_{ij} = \boldsymbol{w}_i^\top (\boldsymbol{C}_1 - \boldsymbol{C}_2) \boldsymbol{w}_j. \tag{A8}$$

**Step 2 (DLIF case)**

For DLIF neurons, we have

$$
\begin{aligned}
(\boldsymbol{Q}_c^{\text{DLIF}})_{ij} &= \mathbb{E}_{\boldsymbol{S} \sim \mathcal{D}_c}[\boldsymbol{I}_{i,:}^{\text{DLIF}}(\boldsymbol{I}_{j,:}^{\text{DLIF}})^T] \\
&= \mathbb{E}_{\boldsymbol{S} \sim \mathcal{D}_c}[(\boldsymbol{w}_i^T \boldsymbol{S} \boldsymbol{S}^T \boldsymbol{w}_j + \Phi_i(\boldsymbol{S}) \boldsymbol{S}^T \boldsymbol{w}_j + \boldsymbol{w}_i^T \boldsymbol{S} \Phi_j^T(\boldsymbol{S}) + \Phi_i(\boldsymbol{S}) \Phi_j^T(\boldsymbol{S}))]
\end{aligned} \tag{A9}
$$

We abbreviate

$$
\begin{aligned}
G_{ij}(c) &:= \mathbb{E}_{\boldsymbol{S} \sim \mathcal{D}_c}\big[\Phi_i(\boldsymbol{S})\boldsymbol{S}^T \boldsymbol{w}_j + \boldsymbol{w}_i^T \boldsymbol{S}\, \Phi_j(\boldsymbol{S})^T\big], \\
H_{ij}(c) &:= \mathbb{E}_{\boldsymbol{S} \sim \mathcal{D}_c}\big[\Phi_i(\boldsymbol{S})\Phi_j(\boldsymbol{S})^T\big].
\end{aligned}
$$

Then

$$
\Delta_{ij}^{\text{DLIF}} = (\boldsymbol{Q}_1^{\text{DLIF}} - \boldsymbol{Q}_2^{\text{DLIF}})_{ij} = \underbrace{\mathbb{E}_{\boldsymbol{S} \sim \mathcal{D}_1}[\boldsymbol{w}_i^T \boldsymbol{S} \boldsymbol{S}^T \boldsymbol{w}_j] - \mathbb{E}_{\boldsymbol{S} \sim \mathcal{D}_2}[\boldsymbol{w}_i^T \boldsymbol{S} \boldsymbol{S}^T \boldsymbol{w}_j]}_{\Delta_{ij}^{\text{LIF}}}
$$
$$
+ \underbrace{G_{ij}(1) - G_{ij}(2)}_{\text{linear-bilinear term}} + \underbrace{H_{ij}(1) - H_{ij}(2)}_{\text{biliear-bilinear term}}. \tag{A10}
$$

Since $s_i(t) \in \{0, 1\}$, we have $s_i(t)^r = s_i(t)$ for any $r \in \mathbb{N}^*$. We can define $s_i(t)s_j(t)$ as the second-order moments for $i \neq j$, $s_i(t)s_j(t)s_k(t)$ as the third-order moments for $i \neq j \neq k$ and so on.

For the linear-bilinear term, we have

$$G_{ij}(c) = \sum_{t=1}^{\tau} \sum_{p \neq q} \sum_u (\boldsymbol{K}_i)_{pq} s_p(t) s_q(t) \boldsymbol{w}_{ju} s_u(t) + \sum_{t=1}^{\tau} \sum_{p \neq q} \sum_u (\boldsymbol{K}_j)_{pq} s_p(t) s_q(t) \boldsymbol{w}_{iu} s_u(t) \tag{A11}$$

Since the higher-order moments are negligible compared to second-order moments, then we have

$$
\begin{aligned}
G_{ij}(c) &= \sum_{p \neq q} [((\boldsymbol{K}_i)_{pq} \boldsymbol{w}_{jp} + (\boldsymbol{K}_i)_{pq} \boldsymbol{w}_{jq} + (\boldsymbol{K}_j)_{pq} \boldsymbol{w}_{ip} + (\boldsymbol{K}_j)_{pq} \boldsymbol{w}_{iq})(\sum_{t=1}^{\tau} s_p(t) s_q(t))] \\
&= \boldsymbol{1}^T \{ [(\boldsymbol{w}_i \boldsymbol{1}^T + \boldsymbol{1} \boldsymbol{w}_i^T) \odot \boldsymbol{K}_j + (\boldsymbol{w}_j \boldsymbol{1}^T + \boldsymbol{1} \boldsymbol{w}_j^T) \odot \boldsymbol{K}_i] \odot \boldsymbol{C}_c \} \boldsymbol{1}
\end{aligned}
\tag{A12}
$$

where $\odot$ is the Hadamard product. And we have

$$
\Delta_{ij}^G = G_{ij}(1) - G_{ij}(2) = \boldsymbol{1}^T \{ [(\boldsymbol{w}_i \boldsymbol{1}^T + \boldsymbol{1} \boldsymbol{w}_i^T) \odot \boldsymbol{K}_j + (\boldsymbol{w}_j \boldsymbol{1}^T + \boldsymbol{1} \boldsymbol{w}_j^T) \odot \boldsymbol{K}_i] \odot (\boldsymbol{C}_1 - \boldsymbol{C}_2) \} \boldsymbol{1} \tag{A13}
$$

Similarly, for the biliear-bilinear term, we have

$$
H_{ij}(c) = \sum_{t=1}^{\tau} \sum_{p \neq q} \sum_{u \neq v} (\boldsymbol{K}_i)_{pq} s_p(t) s_q(t) (\boldsymbol{K}_j)_{uv} s_u(t) s_v(t) \tag{A14}
$$

Since the higher-order moments are negligible compared to second-order moments, then we have

$$
\begin{aligned}
H_{ij}(c) &= \sum_{p \neq q} [(\boldsymbol{K}_i)_{pq} (\boldsymbol{K}_j)_{pq} \sum_{t=1}^{\tau} s_p(t) s_q(t)] \\
&= \boldsymbol{1}^T (\boldsymbol{K}_i \odot \boldsymbol{K}_j \odot \boldsymbol{C}_c) \boldsymbol{1}
\end{aligned}
\tag{A15}
$$

$$
\Delta_{ij}^H = H_{ij}(1) - H_{ij}(2) = \boldsymbol{1}^T [\boldsymbol{K}_i \odot \boldsymbol{K}_j \odot (\boldsymbol{C}_1 - \boldsymbol{C}_2)] \boldsymbol{1} \tag{A16}
$$

**Step 3 (Norm comparison)**

Since
$$
\Delta_{ij}^{\text{DLIF}} = \Delta_{ij}^{\text{LIF}} + \Delta_{ij}^G + \Delta_{ij}^H \tag{A17}
$$
WLOG, we suppose that $\Delta_{ij}^{\text{LIF}} \geq 0$. Since $\|\boldsymbol{C}_1 - \boldsymbol{C}_2\| = 1$, there exists $p \neq q$ such that $k = (\boldsymbol{C}_1 - \boldsymbol{C}_2)_{pq} \neq 0$. We can select $\boldsymbol{K}_i = \boldsymbol{K}_j = \alpha(\mathbf{e}_p \mathbf{e}_q^T + \mathbf{e}_q \mathbf{e}_p^T)$, then

$$
\begin{aligned}
\Delta_{ij}^H &= 2\alpha^2 k \\
\Delta_{ij}^G &= (\boldsymbol{w}_{ip} + \boldsymbol{w}_{iq} + \boldsymbol{w}_{jp} + \boldsymbol{w}_{iq}) \alpha k
\end{aligned}
\tag{A18}
$$

Since $\alpha = 0$ is one of the zeros of $\Delta_{ij}^H + \Delta_{ij}^G$, there exists a small $\epsilon$ such that $\Delta_{ij}^H + \Delta_{ij}^G > 0$ when $\alpha = \epsilon$ and satisfying $\|\boldsymbol{K}_i\|_F = \|\boldsymbol{K}_j\|_F < 1$. Therefore, we have

$$
|\Delta_{ij}^{\text{DLIF}}| = |\Delta_{ij}^{\text{LIF}} + \Delta_{ij}^G + \Delta_{ij}^H| \geq C > 0 \tag{A19}
$$

and

$$
|\Delta_{ij}^{\text{DLIF}}| > |\Delta_{ij}^{\text{LIF}}| \tag{A20}
$$

$$
|\Delta_{ij}^{\text{DLIF}}| = |\Delta_{ij}^{\text{LIF}} + \Delta_{ij}^G + \Delta_{ij}^H| > 0 \tag{A21}
$$

When $\Delta_{ij}^{\text{LIF}} < 0$, we can select $\boldsymbol{K}_i = \alpha(\mathbf{e}_p \mathbf{e}_q^T + \mathbf{e}_q \mathbf{e}_p^T)$ and $\boldsymbol{K}_j = -\alpha(\mathbf{e}_p \mathbf{e}_q^T + \mathbf{e}_q \mathbf{e}_p^T)$, then

$$
\begin{aligned}
\Delta_{ij}^H &= -2\alpha^2 k \\
\Delta_{ij}^G &= (-\boldsymbol{w}_{ip} - \boldsymbol{w}_{iq} + \boldsymbol{w}_{jp} + \boldsymbol{w}_{iq}) \alpha k
\end{aligned}
\tag{A22}
$$

Similarly, since $\alpha = 0$ is still one of the zeros of $\Delta_{ij}^H + \Delta_{ij}^G$, there exists a small $\epsilon$ such that $\Delta_{ij}^H + \Delta_{ij}^G < 0$ when $\alpha = \epsilon$ and satisfying $\|\boldsymbol{K}_i\|_F = \|\boldsymbol{K}_j\|_F < 1$. Therefore, we have

$$
|\Delta_{ij}^{\text{DLIF}}| = |\Delta_{ij}^{\text{LIF}} + \Delta_{ij}^G + \Delta_{ij}^H| \geq C > 0 \tag{A23}
$$

and

$$
|\Delta_{ij}^{\text{DLIF}}| > |\Delta_{ij}^{\text{LIF}}| \tag{A24}
$$

$$
|\Delta_{ij}^{\text{DLIF}}| = |\Delta_{ij}^{\text{LIF}} + \Delta_{ij}^G + \Delta_{ij}^H| > 0 \tag{A25}
$$

In general, we have

$$
\|\Delta^{\text{DLIF}}\|_F \geq C > 0. \tag{A26}
$$

and

$$
\|\Delta^{\text{DLIF}}\|_F > \|\Delta^{\text{LIF}}\|_F. \tag{A27}
$$

This completes the proof. $\qquad \square$

The above lemma demonstrates that, compared to LIF neurons, DLIF neurons can better preserve the correlation of the original input spike trains at the current-input level. (De La Rocha et al., 2007) further proved that, for spiking neurons of the form given in Eq. (1), the correlation of the output spike train is positively correlated with that of the input current, leading to the following lemma.

**Lemma A2.** *Consider two spiking neurons defined by Eq.* (1)*. Let the correlation of their input currents be $a$ in the expectation sense. Then, there exists a constant $k > 0$ such that the correlation of their output spike trains, also measured in expectation, satisfies $b = ka$.*

The proof of Lemma A2 can refer to (De La Rocha et al., 2007).

Then we denote the output spike trains of the hidden neurons as $\boldsymbol{Y} = [\boldsymbol{y}(1), \boldsymbol{y}(2), \cdots, \boldsymbol{y}(\tau)] \in \{0,1\}^{M \times \tau}$. For each class, we define the correlation matrix $\boldsymbol{P}_c$ of $\boldsymbol{Y}$ as $\boldsymbol{P}_c = \frac{1}{\tau}\mathbb{E}_{\boldsymbol{S} \sim \mathcal{D}_c}[\boldsymbol{Y}\boldsymbol{Y}^T]$ for $c \in \{1, 2\}$, and denote the resulting matrices for LIF and DLIF neurons by $\boldsymbol{P}_1^{LIF}, \boldsymbol{P}_2^{LIF}, \boldsymbol{P}_1^{DLIF}, \boldsymbol{P}_2^{DLIF}$, respectively. Let $\boldsymbol{W} = [\boldsymbol{w}_1, \ldots, \boldsymbol{w}_M] \in \mathbb{R}^{M \times N}$ be the weight matrix and $\mathbf{K} = [\boldsymbol{K}_1, \ldots, \boldsymbol{K}_M]$ with $\boldsymbol{K}_m \in \mathbb{R}^{N \times N}$ be the bilinear coefficient matrices. We suppose that $\|\boldsymbol{w}_i\|_F \leq 1$ and $\|\boldsymbol{K}_i\|_F \leq 1$ for $i = 1, \cdots, M$. Then we obtain the following theorem

**Theorem A2.** *Let the input spike trains be drawn from distributions $\mathcal{D}_1$ and $\mathcal{D}_2$ defined in Eq. (A1). Then, for any choice of synaptic weight matrices $\boldsymbol{W}^{\mathrm{LIF}}$ and $\boldsymbol{W}^{\mathrm{DLIF}}$, there exists bilinear coefficient matrices $\mathbf{K}$ for the DLIF neurons such that*

$$\|\boldsymbol{P}_1^{DLIF} - \boldsymbol{P}_2^{DLIF}\|_F \geq C > 0.$$

*and*

$$\|\boldsymbol{P}_1^{DLIF} - \boldsymbol{P}_2^{DLIF}\|_F > \|\boldsymbol{P}_1^{LIF} - \boldsymbol{P}_2^{LIF}\|_F.$$

*Proof.* According to Lemma A1 and Lemma A2, we have

$$\|\boldsymbol{P}_1^{DLIF} - \boldsymbol{P}_2^{DLIF}\|_F = k\|\boldsymbol{Q}_1^{DLIF} - \boldsymbol{Q}_2^{DLIF}\|_F \geq C > 0. \tag{A28}$$

$$\begin{aligned}\|\boldsymbol{P}_1^{DLIF} - \boldsymbol{P}_2^{DLIF}\|_F &= k\|\boldsymbol{Q}_1^{DLIF} - \boldsymbol{Q}_2^{DLIF}\|_F \\ &> k\|\boldsymbol{Q}_1^{DIF} - \boldsymbol{Q}_2^{LIF}\|_F = \|\boldsymbol{P}_1^{LIF} - \boldsymbol{P}_2^{LIF}\|_F\end{aligned} \tag{A29}$$

$\square$

## A.2 DETAILS ABOUT SPIKING NEURON MODELS AND NETWORKS

### A.2.1 DYNAMICS

The sub-threshold somatic membrane potential $V(t)$ is governed by:

$$\tau \frac{dV(t)}{dt} = -(V(t) - V_{rest}) + RI(t), \tag{A30}$$

For computational implementation, we set $R$ and the time interval to 1, yielding the discrete dynamics equations:

$$\begin{cases} U[t] &= (1 - \frac{1}{\tau})V[t-1] + \frac{1}{\tau}I[t], \\ S[t] &= H(U[t] - V_{th}), \\ V[t] &= (1 - S[t])U[t] + S[t]V_{rest}. \end{cases} \tag{A31}$$

where $U[t]$ and $V[t]$ represents the membrane potential before and after reset operations, respectively. $t = 1, 2, 3, \cdots, \tau$ denotes the time step, and $\tau$ is the time duration. $H(x)$ is the Heaviside step function. In our experiments, $V_{th}$ and $V_{rest}$ are set to be 1 and 0, respectively. The choice of $\tau$ is specified in the supplementary.

Eq. (A31) defines the update rule for neuronal dynamics in SNNs. Notably, SNN architectures maintain structural parallels with conventional ANNs, permitting direct adaptation of established frameworks such as VGG , ResNet and Transformer through substitution of activation functions with spiking neuron models. Standard operations including convolution and pooling remain fully compatible.

Table 6: Comparison of Training Time and Memory Cost

| Neuron Model | Network Architecture | Training Time | Memory Cost | Accuracy |
|:---:|:---:|:---:|:---:|:---:|
| LIF | ResNet-34 | 1.54h | 20.17G | 70.23% |
| LIF | ResNet-50 | 1.81h | 23.54G | 70.79% |
| DLIF | ResNet-34 | 1.66h | 22.39G | 71.06% |

SNNs support both neuromorphic and static image data processing. While neuromorphic data naturally contains temporal information, static images require conversion to temporal inputs via pixel value encoding (Rueckauer et al., 2017; Diehl & Cook, 2015; Shrestha & Orchard, 2018). The network output is determined by the highest average firing rate among output layer neurons, which corresponds to the predicted class label.

### A.2.2 TRAINING ALGORITHM

Training spiking neural networks faces a fundamental challenge: the non-differentiability of spike sequences, which prevents direct gradient-based optimization. This limitation is overcome by combining Backpropagation Through Time (BPTT) with surrogate gradient methods (Huh & Sejnowski, 2018; Shrestha & Orchard, 2018; Wu et al., 2018; 2019). The non-differentiable term $\frac{\partial S}{\partial V}$ can be approximated by the surrogate functions such as rectangle or triangle functions (Wu et al., 2018): $\frac{\partial S}{\partial V} = \frac{1}{a}\text{sgn}(|V - V_{th}| < \frac{a}{2})$ or $\frac{\partial S}{\partial V} = \frac{1}{a^2}\max(0, a - |V - V_{th}|)$, where $a$ is a hyperparameter and sgn is the sign function. With the surrogate gradient, the gradient-based algorithm can be applied to train SNNs.

### A.3 EXPERIMENTS OF COMPARING TRAINING TIME AND MEMORY COST

We conducted additional measurements comparing the training time and memory overhead of DLIF with the standard LIF-based SNN. To ensure a fair evaluation, we performed experiments on ImageNet using the ResNet-34 architecture trained with the standard BPTT algorithm, with a batch size of 256, running on a single NVIDIA A100 GPU. The reported training time corresponds to the wall-clock time per epoch. The wall-clock training time and memory cost are computed based on the first three epochs of training.

The results in Table 6 show that DLIF increases per-epoch training time by 8% and GPU memory consumption by 11%. However, this overhead remains acceptable for large-scale SNN training. To further demonstrate that the performance improvement of DLIF is not merely due to an increase in parameter count, we also conducted experiments using the LIF model with ResNet-50. In this setting, the training time and memory consumption are both higher than those of the DLIF-based ResNet-34, yet the accuracy remains lower than that achieved by DLIF-ResNet-34.

### A.4 EXPERIMENTS IN LARGE-SCALE SNNS FOR REINFORCEMENT LEARNING TASKS

Reinforcement learning represents a fundamental machine learning paradigm where agents learn optimal decision-making policies through environmental interactions, guided by reward signals (Kaelbling et al., 1996; Wiering & Van Otterlo, 2012). The Atari game has emerged as a standard benchmark for evaluating RL algorithms, challenging agents to maximize game scores through pixel-level inputs (Mnih et al., 2015). While Q-learning (Watkins & Dayan, 1992) and its deep learning variants have demonstrated strong performance, recent work has successfully adapted spiking neural networks for Q-value approximation, enabling efficient processing of high-dimensional state spaces (Patel et al., 2019; Tan et al., 2021; Chen et al., 2024).

In our experiments, we implement DLIF models within a deep spiking Q-network (DSQN) architecture and evaluate the performance across 16 Atari games (Chen et al., 2024). A comparative analysis with the LIF-based DSQN reveals that DLIF models achieve an average performance improvement of 20.62% in final game scores across all tested environments as shown in Table 7. Due to slight variations in input and output dimensions across different games, the computational cost exhibits minor fluctuations. On average, the DSQN model based on the LIF neuron has $0.21M$ parameters, $0.83M$ FLOPs, and an energy cost of $0.81\mu J$. In comparison, the DLIF-based DSQN incurs $0.23M$

Table 7: Results on Atari Games

| Game | DSQN with LIF (Mean± Std) | DSQN with DLIF (Mean± Std) |
|---|---|---|
| Atlantis | 2515926.7 ± 73782.9 | **2942971.5±73462.3** |
| Beam Rider | 5327.6 ± 178.0 | **7306.9±544.6** |
| Boxing | 82.7 ± 8.7 | **97.0±10.5** |
| Breakout | 368.1 ± 9.8 | **472.1±8.7** |
| Crazy Climber | 95164.4 ± 1232.9 | **100673.6±1519.2** |
| Gopher | 4233.1 ± 176.5 | **6151.4±219.4** |
| Jamesbond | 469.4 ± 82.7 | **587.2±84.7** |
| Kangaroo | 5824.4 ± 540.8 | **7374.2±702.9** |
| Krull | 6991.1 ± 107.0 | **7883.5±262.1** |
| Name this game | 6981.0 ± 192.8 | **8041.2±208.4** |
| Pong | 19.5 ± 0.4 | **20.3±0.6** |
| Road Runner | 27725.6 ± 3954.0 | **29401.1±4507.1** |
| Space Invaders | 1209.9 ± 61.2 | **2769.6±105.3** |
| Star Gunner | 1657.8 ± 102.0 | **1984.4±136.1** |
| Tutankham | 266.5 ± 12.2 | **294.2±11.5** |
| Video Pinball | 408032.6 ± 41687.8 | **436980.0±41047.2** |

Bold values represent the best results for each Atari game

Table 8: Ablation Study of Bilinear Coefficients

| Dataset | Method | Network | Neuron | Mean± Std(%) |
|---|---|---|---|---|
| ImageNet | TET | ResNet-34 | LIF | 64.79 |
| | | | DLIF | **67.32±0.39** |
| | | | DLIF* | 25.83±2.67 |
| | STBP-tdBN | ResNet-34 | LIF | 63.72 |
| | | | DLIF | **66.79±0.53** |
| | | | DLIF* | 21.36±1.89 |
| | Meta-SpikeFormer | Transformer | LIF | 80.00 |
| | | | DLIF | **80.57±0.28** |
| | | | DLIF* | 17.12±5.81 |

Bold values represent the best results for each method;∗ means removing $K$ after training

parameters ($+0.02M, +9.52\%$), $0.9M$ FLOPs ($+0.07M, +8.43\%$), and an energy cost of $0.86\mu J$ ($+0.05\mu J, +6.17\%$). These modest increases in computational and energy costs are justified by the corresponding performance gains.

## A.5 ABLATION STUDY

To highlight the contribution of the bilinear coefficients $K$ in the DLIF model, we perform ablation studies on the ImageNet dataset under three training paradigms: TET, STBP-tdBN, and Meta-SpikeFormer. We consider two ways of removing the bilinear matrices: (i) removing $K$ before training, which reduces the DLIF model to a standard LIF model, and (ii) removing $K$ after training to assess their impact on the learned representations. As shown in Table 8, both settings consistently lead to a drop in test accuracy across all paradigms, confirming the critical role of bilinear coefficients in the DLIF model.

To further examine how different parameterizations of the bilinear matrix $K$ influence DLIF performance, we evaluated two structured variants—a diagonal-block sparse matrix and a low-rank factorization, motivated by biological or computational considerations. In addition to the random 90% sparse matrix used in our main experiments, we constructed (i) a diagonal-block matrix with bandwidth $n = 8$, matched to the same overall sparsity level ( 90%), and (ii) a low-rank factorization $K = UV^T$ with ranks $r = 1, 2, 3$. We also include the zero-rank case ($r = 0$), which removes $K$

Table 9: Comparison of different parameterizations of the bilinear matrix

| Matrix Type | ZR | LR (r=1) | LR (r=2) | LR (r=3) | DB | RS |
|---|---|---|---|---|---|---|
| #Params (M) | 12.17 | 12.30 | 12.43 | 12.56 | 13.21 | 13.21 |
| ACC (%) | 74.38 | 74.55 | 74.82 | 75.27 | 76.82 | 76.89 |

ZR=zero rank; LR=low rank; DB=diagonal block; RS=random sparse

entirely and reduces DLIF to a standard LIF model. All variants were evaluated on CIFAR-100 using ResNet-18 trained with SLTT.

The results in Table 9 highlight two distinct conclusions. First, low-rank parameterization substantially reduces the number of bilinear parameters but yields noticeably weaker performance, with accuracy consistently below the sparse formulations. Second, the diagonal-block matrix performs comparably to the random 90% sparse matrix, suggesting that incorporating locality structure neither improves nor degrades performance under similar sparsity levels. These observations indicate that while our current sparse formulation provides an effective balance between accuracy and parameter efficiency, exploring richer biologically inspired structural priors for $K$ remains an interesting direction for future investigation.

### A.6 IMPLEMENTATION DETAILS

#### A.6.1 PARAMETER COUNT, FLOPS, ENERGY COST

In addition to reporting accuracy, we also evaluate models using three standard metrics: parameter count, FLOPs, and theoretical energy consumption. The parameter count is obtained by summing all trainable weights in the network. When computing the parameter count, we explicitly account for the sparsity of the bilinear matrix—only the non-zero entries after masking are included as trainable parameters. To enforce this sparsity in practice, we generate a fixed binary mask $\mathbf{M}$ before training, where 90% of the entries are set to zero and the remaining 10% to one. During training, the bilinear matrix is parameterized as $\tilde{\mathbf{K}} = \mathbf{K} \odot \mathbf{M}$ where $\odot$ denotes the Hadamard product. This procedure ensures that the effective bilinear parameters strictly follow the desired sparsity pattern, and only the unmasked entries contribute to the parameter count and optimization. For the ResNet and VGG architectures, we incorporate the bilinear operation into the convolutional layers of the first two blocks. For the Transformer architecture, we apply the bilinear operation to the FFN layers in the first two encoder blocks.

FLOPs are estimated by counting all multiplication and addition operations required in a single forward pass across all layers. Theoretical energy consumption is then computed as the weighted sum of these operations, where each multiplication incurs an energy cost of $E_{\text{MAC}}$ and each addition incurs an energy cost of $E_{\text{AC}}$. Following prior work(Kundu et al., 2021; Lemaire et al., 2022; Yao et al., 2025; Zhou et al., 2023; Yao et al., 2023b; Hu et al., 2024), we adopt $E_{\text{MAC}} = 4.6$ pJ and $E_{\text{AC}} = 0.9$ pJ under 45nm CMOS technology(Horowitz, 2014). Accordingly, the total theoretical energy of an SNN is given by $E_{\text{SNN}} = \tau \cdot r \cdot (E_{\text{MAC}} \cdot N_{\text{MAC}} + E_{\text{AC}} \cdot N_{\text{AC}})$, where $N_{\text{MAC}}$ and $N_{\text{AC}}$ denote the total number of multiplications and additions during one forward pass, $\tau$ is the number of timesteps, and $r$ is the average spiking firing rate.

#### A.6.2 OTHER SPIKING NEURON MODELS

The dynamics of the PLIF, GLIF and DH-LIF models can be found in (Fang et al., 2021), (Yao et al., 2022) and citepzheng2024temporal, respectively. The dynamics of the QIF model are defined as:

$$\tau \frac{dv}{dt} = a_0(v - v_{rest})(v - v_c) + RI \tag{A32}$$

while the dynamics of the EIF model are given by:

$$\tau \frac{dv}{dt} = -(v - v_{rest}) + \Delta_T \exp\left(\frac{v - v_{th}}{\Delta_T}\right) + RI \tag{A33}$$

The QIF and EIF models are configured with the following hyperparameters: $v_{rest} = 0$, $v_c = 0.8$, $a_0 = 1$, and $\Delta_T = 1$. For a fair comparison, we evaluate the performance of QIF, EIF, and DLIF

models across CIFAR-10, CIFAR-100, ImageNet, and DVS-CIFAR10 using the same network architecture and hyperparameters as those adopted in (Yao et al., 2022), while for DVS-Gesture as those adopted in (Fang et al., 2021). In addition, when comparing DLIF with the DH-LIF model, we follow the feedforward network architecture with comparable parameters and hyperparameter configurations provided in (Zheng et al., 2024).

### A.6.3 DATASETS

We conduct experiments on a custom-designed XOR task and several visual classification datasets, including CIFAR-10 (Krizhevsky et al., 2009), CIFAR-100 (Krizhevsky et al., 2009), ImageNet (Deng et al., 2009), DVS-Gesture (Amir et al., 2017), and DVS-CIFAR10 (Li et al., 2017). Additionally, we evaluate performance on Atari games using the Gym (Brockman et al., 2016) and CleanRL (Huang et al., 2022) libraries.

**XOR Task**  In the XOR task, each neuron receives 10 Poisson-distributed synaptic inputs over a duration of 200 time steps, with an Poisson rate of 0.5. For inputs where only half of the synapses are activated, the target output spike count is 2. In contrast, for inputs where all synapses are activated, the target output spike count is 0. A total of 1000 samples are generated for training and 200 for testing.

**2D Correlated Input**  We randomly sample two-dimensional Poisson inputs. In each trial, both input dimensions share the same Poisson rate, which is a random number between 0 and 1. Each sequence has a duration of 1000 time steps, and we collect a total of 1200 samples, with 1000 used for training and 200 for testing.

**10D Correlated Input**  We first fix two symmetric input second-order moment matrices, $C_1$ and $C_2$, such that $\|C_1 - C_2\|_F = 1$. The average number of input spikes is set to 200, with a sequence length of 1000. Based on the specified second-order moments and firing rate, we then generate two distinct classes of inputs, with 500 samples per class for training and 100 samples per class for testing.

**CIFAR-10**  The CIFAR-10 dataset comprises 60,000 natural images distributed across 10 classes, with 6,000 images per class. Of these, 50,000 images are designated for training and 10,000 for testing. The dataset is normalized to scale pixel values to the range [0, 1]. Subsequently, each image is replicated $T$ times to generate temporal inputs, where $T$ denotes the number of time steps in the evolution of the SNN.

**CIFAR-100**  The CIFAR-100 dataset is similar to CIFAR-10 but contains 100 classes of objects. It comprises 50,000 training samples and 10,000 test samples. The same preprocessing steps applied to CIFAR-10 are utilized for this dataset.

**ImageNet**  The ImageNet dataset contains over 14 million natural images across more than 20,000 classes, making it one of the largest and most diverse image classification datasets. In this work, we utilize the ImageNet-1K subset, which includes approximately 1.2 million training images and 50,000 validation images across 1,000 classes. Our data pre-processing uses the usual practice, which randomly crops and flips the 224×224 image with general normalization method. Each image is then replicated $T$ times to generate temporal inputs.

**DVS-Gesture**  The DVS-Gesture dataset comprises 1,176 neuromorphic spiking gesture training samples and 288 test samples, each represented as 128×128 pixel frames. The dataset includes 11 gesture types, such as waving, arm rotation, and forearm rolling. It is collected using a dynamic vision sensor, capturing data from 29 subjects under three distinct lighting conditions: natural light, fluorescent light, and LED.

**DVS-CIFAR10**  The DVS-CIFAR10 dataset is derived from the original CIFAR-10 dataset using a neuromorphic vision sensor to generate temporal sequences. It includes 9,000 training sequences and 1,000 test sequences.

**SHD and SSC**  Spiking Heidelberg digits (SHD) and spiking speech command (SSC) datasets convert the original audio data into the spike format through a bionic inner ear model. SHD contains about 10,000 high-quality recordings of English and German speech for digits ranging from 0 to 9. In particular, the SHD training and testing sets contain 8,156 and 2,264 pieces of data, respectively; the SSC training, testing, and validation sets contain 75,466, 9,981, and 20,382 pieces of data.

**Atari Games**  The Atari game dataset comprises data generated from playing various Atari 2600 games, often used for reinforcement learning research. For this study, we evaluate our model using 16 selected games: Atlantis, Beam Rider, Boxing, Breakout, Crazy Climber, Gopher, James Bond, Kangaroo, Krull, Name This Game, Pong, Road Runner, Space Invaders, Star Gunner, Tutankham, and Video Pinball.

### A.6.4 TRAINING METHODS AND SNN ARCHITECTURES

**XOR Task**  For the XOR task, we employ a single DLIF or LIF model. The time constant is set to 2, and the firing threshold is fixed at $V_{th} = 1$. We control the bilinear matrices to be symmetric with zero diagonals and initialize both the weights and bilinear coefficient from a normal distribution. Both DLIF and LIF neurons are trained using the BPTT algorithm, with the mean squared error as the loss function. Training is performed for 100 epochs with a batch size of 128 and a learning rate of 0.1. We constrain the Frobenius norm of $K$ to be 1 during training.

**Network Correlation Simulation**  For the two-dimensional case, we train DLIF and LIF models to maximize the normalized correlation of the output spike trains toward 1. For the ten-dimensional case, we train the models such that the Frobenius norm of the difference between the output correlation matrices of the two input classes approaches 1. In both settings, the mean squared error is used as the loss function. The bilinear matrices are constrained to be symmetric with zero diagonals, and both the weights and bilinear coefficients are initialized from a normal distribution. Training is conducted using the BPTT algorithm for 100 epochs with a batch size of 128 and a learning rate of 0.1. During training, we constrain the Frobenius norms of both the weight vectors and the bilinear matrices to be 1.

**Experiments in Large-scale SNNs**  In the results presented in Section 4, we consistently control the bilinear matrices to be symmetric with zero diagonals, and, except for the ablation experiments, we always fix the sparsity level at 90%. For all experiments, the training process is repeated five times with different initial values, and the mean and standard deviation of test set accuracy are calculated. For experiments where the oringcal spiking neurons are replaced with DLIF neurons in SNNs, we adopt the same training methods and network architectures as described in the original works, which are detailed as below:

**SLTT**  Spatial Learning Through Time (SLTT) (Meng et al., 2023) is a training method designed to reduce the number of scalar multiplications and achieve memory efficiency that is independent of the total number of time steps, compared with BPTT. For CIFAR-10, CIFAR-100, DVS-Gesture, and DVS-CIFAR10, the adopted network architectures are ResNet-18, ResNet-18, VGG-11, and VGG-11, respectively, with the total number of time steps set to 6, 6, 20, and 10, respectively. The time constant is set to $\tau = 2$. To improve compatibility with neuromorphic hardware, max-pooling layers are replaced with average-pooling layers in the network architectures. The loss function combines cross-entropy loss and mean-squared-error loss (Deng et al., 2022). For all tasks, stochastic gradient descent (SGD) (Rumelhart et al., 1986) with a momentum of 0.9 is employed to train the networks, and a cosine annealing scheduler (Loshchilov & Hutter, 2016) is used to adjust the learning rate. The training hyperparameters are as follows: for the CIFAR-10 and CIFAR-100 datasets, models are trained for 200 epochs with a learning rate of 0.1, a batch size of 128, and weight decays of $5 \times 10^{-5}$ and $5 \times 10^{-4}$, respectively. For the DVS-Gesture dataset, models are trained for 300 epochs with a learning rate of 0.1, a batch size of 16, and a weight decay of $5 \times 10^{-4}$. For the DVS-CIFAR10 dataset, models are trained for 300 epochs with a learning rate of 0.05, a batch size of 128, and a weight decay of $5 \times 10^{-4}$.

**ESG**  Evolutionary Surrogate Gradients (ESG) (Guo et al., 2022a) is a novel method for differentiable spike activity estimation, designed to ensure sufficient model updates during the initial stages

of training and accurate gradient calculations at later stages. For CIFAR-10, CIFAR-100, and DVS-CIFAR10, the adopted network architectures are VGG-16, VGG-16, and ResNet-19, respectively, with the total number of time steps set to 5, 5, and 10, respectively. The time constant is configured as $\tau = 1.33$. To enhance compatibility with neuromorphic hardware, max-pooling layers are replaced with average-pooling layers in the network architectures. Cross-entropy loss is used as the loss function. For all tasks, the SGD optimizer with a momentum of 0.9 is employed to train the networks over 100 epochs, with a batch size of 128, a learning rate of 0.1, and a weight decay of $1 \times 10^{-4}$.

**OTTT** Online Training Through Time (OTTT) (Xiao et al., 2022) is an extension of BPTT that enables forward-in-time learning by tracking presynaptic activities and leveraging instantaneous loss and gradients. The VGG-11 network architecture is utilized for all experiments on CIFAR-10, CIFAR-100, DVS-CIFAR10, and DVS-Gesture, with the total number of time steps set to 6, 6, 20, and 10, respectively. The time constant is configured as $\tau = 2$. All models are trained using the SGD optimizer with a momentum of 0.9. For CIFAR-10, CIFAR-100, and DVS-CIFAR10, models are trained for 300 epochs with a batch size of 128. The initial learning rate is set to 0.1 and decayed to 0 using a cosine annealing scheduler. For DVS-CIFAR10, dropout is applied to all layers with a dropout rate of 0.1. The loss function combines cross-entropy loss and mean-squared-error loss. For DVS-Gesture, models are trained for 300 epochs with a batch size of 16 using the Adam optimizer. The initial learning rate is set to 0.001 and decayed to 0 using a cosine annealing scheduler.

**STBP-tdBN** The threshold-dependent batch normalization method based on spatio-temporal back-propagation (STBP-tdBN) (Zheng et al., 2021) addresses the gradient vanishing and explosion problems while adjusting the firing rate. For CIFAR-10, ImageNet, DVS-Gesture, and DVS-CIFAR10, the adopted network architectures are ResNet-19, ResNet-34, ResNet-17, and ResNet-19, respectively, with the total number of time steps set to 6, 6, 40, and 10, respectively. The time constant is configured as $\tau = 1.33$. Cross-entropy loss is used as the loss function. For all experiments, the SGD optimizer is employed with an initial learning rate of 0.1, a momentum of 0.9, a batch size of 40, and 500 training epochs. The learning rate $r$ decays to $0.1r$ every 35 epochs.

**TET** The Temporal Efficient Training (TET) algorithm (Deng et al., 2022) is designed to mitigate the loss of momentum in gradient descent with stochastic gradients (SG), facilitating convergence to flatter minima and improved generalizability. The ResNet-34 architecture is utilized for the ImageNet dataset, with the time constant configured as $\tau = 2$. The SGD optimizer with a momentum of 0.9 and a weight decay of $4 \times 10^{-5}$ is employed. The learning rate is initialized at 0.1 and decays to 0 using a cosine schedule. The network is trained for 120 epochs.

**SSNN** The Shrinking Spiking Neural Network (SSNN) (Ding et al., 2024) is designed to achieve low-latency neuromorphic object recognition. The VGG-9 network architecture is utilized with a total of 8 time steps across all datasets. The time constant is configured as $\tau = 2$. All models are trained for 100 epochs with an initial learning rate of 0.1, which decays to one-tenth of its previous value every 30 epochs. The batch size is set to 64, and the SGD optimizer with a momentum of 0.9 and a weight decay of $1 \times 10^{-3}$ is employed.

**ESL** The Error Compensation Learning (ESL) (Liu et al., 2025a) introduced a learnable threshold clipping function, dual-threshold neurons, and an optimized membrane potential initialization strategy to mitigate the conversion error. For CIFAR-10 and ImageNet, the adopted network architectures are ResNet-18 and VGG-16, with the total number of time steps set to 64 and 128. respectively. We use the same training strategy as in STBP-tdBN since the original text does not provide a detailed description of the training parameters.

**TSER** Temporal Separation with Entropy Regularization (TSER) (Yu et al., 2025a) introduced knowledge distillation in spiking neural networks. The teacher models for CIFAR-10, CIFAR and ImageNet datasets are VGG-16, VGG-16 and ResNet-34, respectively. The time steps are all set to 4. We use the same training strategy as in STBP-tdBN since the original text does not provide a detailed description of the training parameters.

**FSTA** Frequency-based Spatial-Temporal Attention (FSTA) module is proposed to enhance feature learning in SNNs (Yu et al., 2025b). For CIFAR-10, CIFAR-100, ImageNet, and DVS-CIFAR10, the

adopted network architectures are ResNet-19, ResNet-19, ResNet-34, and ResNet-20, respectively, with the total number of time steps set to 2, 2, 4, and 16, respectively. We use the same training strategy as in STBP-tdBN since the original text does not provide a detailed description of the training parameters.

**SSSA**    Saccadic Spike Self-Attention (SSSA) method is proposed to address the issues of the mismatch between the vanilla self-attention mechanism and spatio-temporal spike trains (Wang et al., 2025b). ViTs are used for CIFAR-10, CIFAR-100, ImageNet and DVS-CIFAR10 datasets with time steps of 4. We use the same training strategy as in STBP-tdBN since the original text does not provide a detailed description of the training parameters.

**Spike-driven Transformer**    The Spike-driven Transformer is an SNN architecture that incorporates the spike-driven paradigm into Transformer (Yao et al., 2023a). This architecture combines the low power of SNN and the excellent accuracy of the Transformer. The time constant is set to $\tau = 2$. For ImageNet, the batch size is set to 256 during 310 training epochs with a cosine-decay learning rate whose initial value is 0.0005. The optimizer is Lamb and the timestep is $T = 4$. For the other four datasets, we employ timesteps $T = 4$ on CIFAR-10 and CIFAR-100, and $T = 16$ on DVS-CIFAR10 and DVS-Gesture. The training epoch for these four datasets is 200. The batch size is 32 for CIFAR10/100 and 16 for DVS-Gesture/CIFAR10. The learning rate is initialized to 0.0005 for CIFAR10/100, 0.0003 for DVS-Gesture, and 0.01 for DVS-CIFAR10. All of them are reduced with cosine decay. In addition, the network structures used in CIFAR-10, CIFAR-100, ImageNet, DVS-CIFAR10, and DVS-Gesture are: Transformer-2-512, Transformer-2-512, Transformer-10-512 Transformer-2-256 and Transformer-2-256, respectively, where Transformer-$L$-$D$ in represents a model with $L$ encoder blocks and $D$ channels.

**Meta-SpikeFormer**    Meta-SpikeFormer is a general Transformer-based SNN architecture for future next-generation Transformer-based neuromorphic chip designs (Yao et al., 2024). A 55M Transformer is used for ImageNet dataset with 4 timesteps. The AdamW is employed with an initial learning rate of $1e - 3$ that will decay in the polynomial decay schedule with a power of 0.9. To speed up training, we warm up the model for 1.5k iterations with a linear decay schedule.

**DSQN**    The network architecture of Deep Spiking Q-Network (DSQN) (Chen et al., 2024) is structured as Input-32C8S4-SN-64C4S2-SN-64C3S1-SN-Flatten-512-SN-$N_A$-SN, where SN represents spiking neurons, which can be either LIF or DLIF neurons, and $N_A$ denotes the number of actions in the task. The model is trained over a total of 20 million frames. During evaluation, the agent begins each episode with a random number (up to 30) of no-op actions, and the behavior policy follows an $\epsilon$-greedy approach, with $\epsilon$ fixed at 0.05. The simulation timesteps are set to 8, and the membrane time constant is configured as $\tau = 2$. The model uses a minibatch size of 32, a replay start size of 50,000, and a replay memory size of 1,000,000. The target network is updated every 10,000 steps. The Adam optimizer is employed with a learning rate of 0.00025 and an $\epsilon$ value of $1 \times 10^{-8}$. Exploration starts at an initial value of 1.0, decaying linearly to a final value of 0.1 over 1,000,000 frames. The maximum number of no-op actions per episode is set to 30. These hyperparameters remain consistent across all games.

All implementations are built on the PyTorch (Paszke et al., 2019) and SpikingJelly (Fang et al., 2023) frameworks. All the experiments are conducted on an NVIDIA Tesla A100 GPU with 6,912 CUDA cores and 432 tensor cores.

