# OpenReview forum: "Beyond Linear Processing: Dendritic Bilinear Integration in Spiking Neural Networks"
_ICLR.cc/2026/Conference — ICLR 2026 Poster_

### Official Review · Reviewer_VTNi · 2025-10-21

**Soundness:** 3
**Presentation:** 3
**Contribution:** 2
**Rating:** 4
**Confidence:** 5

**Summary:**

This paper introduces the DLIF neuron—a variant of the Leaky-Integrate-and-Fire model that implements dendritic bilinear integration. Compared with the standard LIF, DLIF adds a bilinear dendritic term that can be evaluated with minimal overhead, requiring only AND operations. The authors theoretically prove that a single DLIF unit can discriminate between spike trains with identical mean firing rates but different correlation structures, and that a two-layer DLIF-based SNN preserves input correlation structures more effectively than its LIF counterpart. Extensive experiments on multiple datasets and across many existing architectures or training methods show consistent accuracy gains when LIF neurons are replaced by DLIF.

**Strengths:**

- Theoretical analysis rigorously supports the advantages of DLIF.
- Exhaustive empirical evaluation: DLIF consistently improves performance when plugged into a wide range of models.

**Weaknesses:**

- DLIF appears difficult to realize at the neuron level alone, because it requires storing the presynaptic spike vectors from the previous layer rather than just the aggregated synaptic current. Implementing this seemingly demands stateful synapses, potentially increasing both training time and memory usage.
- The proposed method was evaluated on multiple datasets by replacing the LIF neurons in prior works with DLIF neurons. Tables 1 and 2 show consistent and stable improvements, a result that is highly encouraging and suggests DLIF could universally outperform LIF and become a new standard neuron model for SNNs. However, the absence of released source code and experiment logs reduces credibility and prevents readers from examining implementation details.

**Questions:**

The authors claim that DLIF neurons incur “no significant increase in computational cost”; however, in my experience introducing stateful synapses into an SNN usually raises training expenses dramatically. Could the authors supply a direct comparison of training/inference time and memory consumption between DLIF and vanilla LIF neurons?

---

> ### Author Response · Authors · 2025-11-28
>
> **Weakness1:** DLIF appears difficult to realize at the neuron level alone, because it requires storing the presynaptic spike vectors from the previous layer rather than just the aggregated synaptic current. Implementing this seemingly demands stateful synapses, potentially increasing both training time and memory usage.
>
> **Response**: We thank the reviewer for the comment. Compared with a standard LIF neuron, our DLIF implementation does not introduce any additional neuron-level state. The aggregated synaptic current in both LIF and DLIF models depends on the presynaptic spike vectors from the last time step and do not rely on earlier ones. The difference is  that LIF performs a linear integration, whereas DLIF includes an additional bilinear integration term.
>
> Following the reviewer’s suggestion, we also conducted additional measurements comparing the training time and memory overhead of DLIF with the standard LIF-based SNN. To ensure a fair evaluation, we performed experiments on ImageNet using the ResNet-34 architecture trained with the standard BPTT algorithm, with a batch size of 256, running on a single NVIDIA A100 GPU. The reported training time corresponds to the wall-clock time per epoch. The wall-clock training time and memory cost are calculated based on the first three epochs of training.
>
> | neuron type | training time | memory cost |
> | :---------: | :-----------: | :---------: |
> |    DLIF     |     1.66h     |   22.39G    |
> |     LIF     |     1.54h     |   20.17G    |
>
> The results show that DLIF increases per-epoch training time by 8% and GPU memory consumption by 11%, the overhead remains acceptable for large-scale SNN training.
>
> To further demonstrate that the performance improvement of DLIF is not merely due to an increase in parameter count, we also conducted experiments using the LIF model on ResNet-50. In this setting, the training time and memory consumption are both higher than those of the DLIF-based ResNet-34, yet the accuracy remains lower than that achieved by DLIF-ResNet-34.
>
> | neuron type | network architecture | training time | memory cost | accuracy |
> | :---------: | :------------------: | :-----------: | :---------: | :------: |
> |    DLIF     |      ResNet-34       |     1.66h     |   22.39G    |  71.06%  |
> |     LIF     |      ResNet-50       |     1.81h     |   23.54G    |  70.79%  |
>
> We will include these new experimental results in the revised manuscript.
>
> **Weakness2:** The proposed method was evaluated on multiple datasets by replacing the LIF neurons in prior works with DLIF neurons. Tables 1 and 2 show consistent and stable improvements, a result that is highly encouraging and suggests DLIF could universally outperform LIF and become a new standard neuron model for SNNs. However, the absence of released source code and experiment logs reduces credibility and prevents readers from examining implementation details.
>
> **Response**:  We thank the reviewer for this suggestion. In compliance with the double-blind review policy, we have uploaded the code in the supplementary material. The official GitHub repository will be made publicly available upon acceptance.
>
> **Question1**: The authors claim that DLIF neurons incur “no significant increase in computational cost”; however, in my experience introducing stateful synapses into an SNN usually raises training expenses dramatically. Could the authors supply a direct comparison of training/inference time and memory consumption between DLIF and vanilla LIF neurons?
>
> **Response**:  We kindly refer the reviewer to our response to Weakness 1, where we provide a detailed comparison of the training time and memory usage between DLIF and vanilla LIF neurons. In brief, DLIF increases the per-epoch training time by only 8% and the GPU memory by 11% on ImageNet with ResNet-34, confirming that DLIF does not introduce significant computational or memory overhead. To further show that DLIF’s performance gain is not simply due to an increase in parameter count, we also evaluated an LIF-based ResNet-50. Although it requires more training time and memory than DLIF-ResNet-34, its accuracy remains lower.

---

### Official Review · Reviewer_KEoS · 2025-10-29

**Soundness:** 3
**Presentation:** 3
**Contribution:** 3
**Rating:** 8
**Confidence:** 3

**Summary:**

The paper proposes a DLIF neuron that augments the input current with a bilinear dendritic term ($s^\top K s$), argues theoretically that DLIF captures pairwise input correlations (single-neuron) and preserves/propagates correlation structure (network-level), and reports consistent gains across CIFAR-10/100, ImageNet, DVS-Gesture, DVS-CIFAR10, and Atari RL with small overheads. The idea is timely and simple to integrate into existing SNN stacks. Results are encouraging, but the claims about complexity/energy, the parameterization and sparsity of ($K$), and the scope/limits of the theory need clearer, checkable evidence.

**Strengths:**

- DLIF replaces linear current summation with an additive bilinear term (s^\top K s); formulation is simple, but performance is good.
- Static and neuromorphic vision, multiple architectures (VGG/ResNet/Transformer), and training regimes (SLTT, OTTT, TET, STBP-tdBN, etc.) show consistent improvements with small reported overheads.

**Weaknesses:**

1. The text states ($K$) reflects spatial relationships and is intensity-independent, yet in networks it is learned. Are there structured priors (locality, block-diagonal, cross-channel only) or sharing tied to receptive fields? A comparison of fully learnable ($K$) vs. structured/low-rank/near-neighbor ($K$) would clarify the interpretability–performance–cost trade-off.

2. Since $K \in \mathbb{R}^{N\times I \times I}$ (although sparse), computational cost, parameter accounting, and training speed need to be better compared.

**Questions:**

1. How about replace sparse matrix to low-rank matrix. It can better reduce parameter number and computational cost.

---

> ### Author Response · Authors · 2025-12-03
> **Response to Reviewer KEoS (1/2)**
>
> **Weakness1:** The text states (K) reflects spatial relationships and is intensity-independent, yet in networks it is learned. Are there structured priors (locality, block-diagonal, cross-channel only) or sharing tied to receptive fields? A comparison of fully learnable (K) vs structured/low-rank/near-neighbor (K) would clarify the interpretability–performance–cost trade-off.
>
> **Response**: We thank the reviewer for the insightful suggestion. According to *Li et al., 2019*, the bilinear dendritic interaction matrix in biological neurons is indeed sparse and locally structured. Motivated by this biological prior, we conducted additional experiments to evaluate whether incorporating such structure into the bilinear matrix $K$ can further improve performance or interpretability.
>
> In the main paper, we use the random sparse $K$ to control parameter count and match the sparsity level motivated by biological measurements. To test the reviewer’s suggestion, we implemented a structured variant of $K$ by constraining it to an n-diagonal block matrix (with zero diagonal), where the bandwidth was set to n=8 so that the overall sparsity (~90%) is matched to the random-sparse formulation. We compared this structured K with both the fully learnable K and the random sparse K used in the main paper. Experiments were performed on CIFAR-100 with ResNet-18 trained using SLTT.
>
> | Matrix Structure | diagonal block | full learnable | random sparse |
> | :--------------: | :------------: | :------------: | :-----------: |
> |    #Params(M)    |     13.21      |     22.57      |     13.21     |
> |      ACC(%)      |     76.82      |     78.67      |     76.89     |
>
> The locality-inspired diagonal-block structure did not provide a clear performance advantage over the sparsity-only random formulation. Nevertheless, we agree that incorporating stronger biological priors—such as locality, block structure, or dendritic wiring constraints—remains a promising direction for future exploration. We will include these new experimental results in the revised manuscript and expand the discussion on the potential benefits and limitations of imposing additional structure on the bilinear interaction matrix.
>
> **Weakness2:** Since (although sparse), computational cost, parameter accounting, and training speed need to be better compared.
>
> **Response**: We thank the reviewer for raising this question. Following the reviewer’s suggestion, we performed additional analyses comparing DLIF with standard LIF-based SNNs in terms of training-time overhead, memory usage, parameter accounting, and theoretical computational cost.
>
> - Training-time and memory measurements.
>
> We conducted controlled experiments on ImageNet using the ResNet-34 architecture trained with FSTA, a batch size of 256, and a single NVIDIA A100 GPU. Training time is measured as the wall-clock time per epoch
>
> |      | training time | memory cost |
> | :--: | :-----------: | :---------: |
> | DLIF |     1.66h     |   22.39G    |
> | LIF  |     1.54h     |   20.17G    |
>
> The results show that DLIF increases per-epoch training time by 8% and GPU memory consumption by 11%, the overhead remains well within acceptable limits for large-scale SNN training.
>
> - Parameter accounting
>
> For a convolution layer with $C_{in}$ input channels and $C_{out}$ output channels as an example, a LIF neuron at the $j$th output channel will linearly integrate signals across input channels with $C_{in}\times C_{out}$ parameters.
>
> When introducing DLIF neurons, we consider the second-order interactions between input channels. Because we enforce 90% sparsity on each K, the number of extra learnable parameters contributed by the bilinear term is $0.1\times C_{in}^2\times C_{out}$.
>
> - Computational cost and theoretical energy estimation.
>
> We follow the standard computation methodology of theoretical energy cost  in prior work (Zhou et al., 2023; Yao et al., 2023). The standard way to estimate the theoretical energy consumption of an SNN layer is
> $$
> E=E_{AC}/E_{MAC}\times T \times fr\times FLOPs,
> $$
> where $E_{AC}$ and $E_{MAC}$ represent the energy cost of MAC and AC operation, fr is the average firing rate of the presynaptic spike train of that layer, and T is the number of simulation time steps. When considering the cost of additional bilinear operation in DLIF, note that computing the pairwise product $x_ix_j$ can be implemented using simple logical operations in spike-based hardware and therefore does not introduce extra MAC operations. Consequently, the additional energy cost introduced by the bilinear term is computed as
> $$
> E_{bilinear}=E_{AC}\times T \times fr\times FLOPs_{bilinear},
> $$
> This energy estimate for the bilinear component is consistent with the computation methodology used for standard LIF models (Zhou et al., 2023; Yao et al., 2023).

---

> ### Author Response · Authors · 2025-12-03
> **Response to Reviewer KEoS (2/2)**
>
> **Questions1:** How about replace sparse matrix to low-rank matrix. It can better reduce parameter number and computational cost.
>
> **Response**: We thank the reviewer for the constructive suggestion of replacing the sparse bilinear matrix with a low-rank factorization. Following this suggestion, we conducted additional experiments based on the CIFAR-100 dataset using the ResNet-18 architecture trained with SLTT, where the bilinear coefficient matrix $\mathbf{K}\in\mathbb{R}^{n\times n}$ was replaced by a low-rank decomposition of the form: $\mathbf{K}=\mathbf{U}\mathbf{V}^T$,
>
> where $\mathbf{U},\mathbf{V}\in\mathbb{R}^{n\times r}$, and r is the rank. We evaluated ranks r=1,2,3. This low-rank parameterization can reduce the number of extra learnable parameters from $0.1\times N^2$ to $r\times N$, achieving an even stronger compression than the 90% sparse bilinear matrix used in our main experiments. We also include the zero-rank case ($r=0$), which removes $\mathbf{K}$ entirely and reduces DLIF to a standard LIF model.
>
> | Matrix type | Zero rank | Low rank(r=1) | Low rank(r=2) | Low rank(r=3) | Sparse |
> | :---------: | :------------: | :-----------: | :-----------: | :-----------: | :----: |
> | #Params(M)  |     12.17      |     12.30     |     12.43     |     12.56     | 13.21  |
> |   ACC(%)    |     74.38      |     74.55     |     74.82     |     75.27     | 76.89  |
>
> The results show that low-rank parameterization indeed reduces the number of parameters even further compared to our 90% sparse bilinear matrix. However, the accuracy improvements over the standard LIF model are noticeably smaller than those achieved using the sparse bilinear formulation. .
>
> We will include these new results in the revised manuscript and add a discussion noting that low-rank parameterization is a promising direction for model compression.
>
> **References**
>
> [1] Li, Songting, et al. "Dendritic computations captured by an effective point neuron model." Proceedings of the National Academy of Sciences 116.30 (2019): 15244-15252.
>
> [2] Zhou, Zhaokun, et al. "Spikformer: When Spiking Neural Network Meets Transformer." The Eleventh International Conference on Learning Representations. 2023.
>
> [3] Yao, Man, et al. "Attention spiking neural networks." *IEEE transactions on pattern analysis and machine intelligence* 45.8 (2023): 9393-9410.

---

### Official Review · Reviewer_Uc9p · 2025-10-31

**Soundness:** 3
**Presentation:** 3
**Contribution:** 2
**Rating:** 4
**Confidence:** 4

**Summary:**

The paper proposes the Dendritic LIF (DLIF) model that extends conventional LIF by incorporating bilinear dendritic integration. Through theoretical analysis and empirical simulations, the authors show that a single DLIF neuron can effectively capture correlations in input spike trains. Experiments across diverse deep SNN architectures and datasets demonstrate that DLIF can achieve SOTA performance, highlighting its scalability.

**Strengths:**

1.Introducing dendritic computation to large-scale neuromorphic computing is an advanced and compelling research topic.

2.The paper is clearly structured (from single neuron to large-scale SNNs) and is generally well written. The mathematical derivations are rigorous. The figures and tables effectively convey the key ideas and results.

3.Experiments cover a wide range of tasks and SNN architectures. DLIF consistently outperforms LIF on (almost) all the tasks, showcasing the method’s effectiveness and generalizability.

**Weaknesses:**

1.The novelty of the proposed model is not clearly stated. What are the key differences or innovations of DLIF with the model proposed by (Li et al., 2019)?

2.Although comparisons are made with point neuron models like PLIF, GLIF, QIF and EIF (Table 3), no comparison is made with dendritic or multi-compartment spiking neurons like (Zheng et al., 2024). The authors claim that DLIF is inspired by dendritic computation, so comparing it with other dendritic neuron models would strengthen the work’s soundness.

3.While inference cost metrics like FLOPS and theoretical energy consumption are provided, there is no information regarding training cost. How much extra training time and memory overhead does the bilinear operation introduce? A comparison with LIF or other spiking neuron models should be made.

Li, Songting, et al. "Dendritic computations captured by an effective point neuron model." Proceedings of the National Academy of Sciences 116.30 (2019): 15244-15252.

Zheng, Hanle, et al. "Temporal dendritic heterogeneity incorporated with spiking neural networks for learning multi-timescale dynamics." Nature Communications 15.1 (2024): 277.

**Questions:**

See “Weakness” for major questions. Below are some additional minor questions and suggestions:

1.More details should be provided regarding the computation of theoretical energy cost. The explanation in Appendix cannot address my key concern: how is the energy consumption of a bilinear operation calculated?

2.The parameter counts in Table 1 and 2 are weird for me. For a synapse layer with $N$ presynaptic neurons and $M$ postsynaptic neurons, the linear weight is a $M\times N$ matrix, while the bilinear weight is a $M\times N \times N$ tensor, as stated in Line 192 of the manuscript. Even with a 90% sparsity constraint and an all-zero diagonal, there should still be about $0.1\times M \times (N^2-N)$ more learnable parameters, which is much larger than $M \times N$. However, the reported parameter increase is only about 10%. Please clarify in detail how the parameter counts are computed.

3.The authors report the results of DLIF networks trained with OTTT. However, OTTT is an eligibility-trace-based online learning method designed for optimizing $\mathbf{W}$, the linear weights. Its extension to bilinear weights is not trivial. Please explain how to use OTTT to optimize bilinear weights.

4.Typos: “biliear” in Related Work should be bilinear; “as a binary” in Page 3, Line 159 should be “as a binary matrix”.

5.Citation formatting: please use `\citep{}` instead of `(\cite{})`. The displayed text should appear as (xxx et al., 20xx) instead of (xxx et al., (20xx)).

---

> ### Author Response · Authors · 2025-11-28
> **Response to Reviewer Uc9p (1/3)**
>
> **Weakness1:** The novelty of the proposed model is not clearly stated. What are the key differences or innovations of DLIF with the model proposed by (Li et al., 2019)?
>
> **Response**: We thank the reviewer for the question. Li et al. (2019) proposed a conductance-based model that captures bilinear dendritic integration but cannot be used directly within an SNN framework. Conductance-based synapses introduce voltage-dependent, multiplicative dynamics and per-synapse conductance states, which substantially increase computational and memory complexity, making them difficult to scale to deep SNN architectures. Our contribution is to adapt this dendritic bilinear mechanism into a current-based formulation that is fully compatible with modern SNN architectures, efficient to compute, and amenable to gradient-based training, while retaining the biologically inspired bilinear rule.
>
> Please note that, as stated in the third paragraph of the Introduction and the beginning of Section 3.1 in the original submission, our work was explicitly inspired by Li et al. (2019). We do not overstate our contribution nor overlook theirs.  We will include an expanded discussion of these differences in the revised manuscript.
>
> **Weakness2:** Although comparisons are made with point neuron models like PLIF, GLIF, QIF and EIF (Table 3), no comparison is made with dendritic or multi-compartment spiking neurons like (Zheng et al., 2024). The authors claim that DLIF is inspired by dendritic computation, so comparing it with other dendritic neuron models would strengthen the work’s soundness.
>
> **Response**: We thank the reviewer for the insightful suggestion. Following your comment, we conducted an additional comparison with the dendritic neuron model proposed by Zheng et al. (2024), namely the LIF neuron model with temporal dendritic heterogeneity (DH-LIF). Our experimental results show that DLIF consistently outperforms the DH-LIF model over two spiking speech recognition tasks, SHD and SSC with comparable parameters. （We chose to conduct comparisons on the SHD and SSC datasets because Zheng et al. (2024) did not include experiments on the datasets used in our study, namely CIFAR-10, CIFAR-100, ImageNet, DVS-Gesture, and DVS-CIFAR10.）
>
> We will include the results in the revised manuscript.
>
> |        |  SHD   |  SSC   |
> | :----: | :----: | :----: |
> |  DLIF  | 92.71% | 83.13% |
> | DH-LIF | 92.10% | 82.46% |
>
> **Weakness3**: While inference cost metrics like FLOPS and theoretical energy consumption are provided, there is no information regarding training cost. How much extra training time and memory overhead does the bilinear operation introduce? A comparison with LIF or other spiking neuron models should be made.
>
> **Response**: We thank the reviewer for raising this point. Following the reviewer’s suggestion, we conducted additional measurements comparing the training time and memory overhead of DLIF with the standard LIF-based SNN. To ensure a fair evaluation, we performed experiments on ImageNet using the ResNet-34 architecture trained with the standard BPTT algorithm, with a batch size of 256, running on a single NVIDIA A100 GPU. The reported training time corresponds to the wall-clock time per epoch. The wall-clock training time and memory cost are calculated based on the first three epochs of training.
>
> | neuron type | training time | memory cost |
> | :---------: | :-----------: | :---------: |
> |    DLIF     |     1.66h     |   22.39G    |
> |     LIF     |     1.54h     |   20.17G    |
>
> The results show that DLIF increases per-epoch training time by 8% and GPU memory consumption by 11%, the overhead remains acceptable for large-scale SNN training.
>
> To further demonstrate that the performance improvement of DLIF is not merely due to an increase in parameter count, we also conducted experiments using the LIF model on ResNet-50. In this setting, the training time and memory consumption are both higher than those of the DLIF-based ResNet-34, yet the accuracy remains lower than that achieved by DLIF-ResNet-34.
>
> | neuron type | network architecture | training time | memory cost | accuracy |
> | :---------: | :------------------: | :-----------: | :---------: | :------: |
> |    DLIF     |      ResNet-34       |     1.66h     |   22.39G    |  71.06%  |
> |     LIF     |      ResNet-50       |     1.81h     |   23.54G    |  70.79%  |
>
> We will include these new experimental results in the revised manuscript.

---

> > ### Author Response · Authors · 2025-11-28
> > **Response to Reviewer Uc9p (2/3)**
> >
> > **Question1:** More details should be provided regarding the computation of theoretical energy cost. The explanation in Appendix cannot address my key concern: how is the energy consumption of a bilinear operation calculated?
> >
> > **Response**: We follow the standard methodology for estimating theoretical energy cost in prior work (Zhou et al., 2023; Yao et al., 2023). The standard way to estimate the theoretical energy consumption of an SNN layer is
> > $$
> > E=E_{AC}/E_{MAC}\times T \times \text{fr} \times\text{FLOPs},
> > $$
> > where $E_{AC}$ and $E_{MAC}$ represent the energy cost of MAC and AC operations, fr is the average firing rate of the presynaptic spike train of that layer, and T is the number of simulation time steps. When considering the cost of additional bilinear operation in DLIF, note that computing the pairwise product $x_ix_j$ can be implemented using simple logical operations in spike-based hardware and therefore does not introduce extra MAC operations. Consequently, the additional energy cost introduced by the bilinear term is computed as
> > $$
> > E_{bilinear}=E_{AC}\times T \times fr*FLOPs_{bilinear},
> > $$
> > This energy estimate for the bilinear component is fully consistent with the computation methodology used for standard LIF models (Zhou et al., 2023; Yao et al., 2023). We will include the details of the theoretical energy calculation in the revised manuscript.
> >
> > **Question2:** The parameter counts in Table 1 and 2 are weird for me. For a synapse layer with $N$ presynaptic neurons and $M$ postsynaptic neurons, the linear weight is a $M\times N$ matrix, while the bilinear weight is a $M\times N \times N$ tensor, as stated in Line 192 of the manuscript. Even with a 90% sparsity constraint and an all-zero diagonal, there should still be about $0.1\times M \times (N^2-N)$ more learnable parameters, which is much larger than $M \times N$. However, the reported parameter increase is only about 10%. Please clarify in detail how the parameter counts are computed.
> >
> > **Response:** We thank the reviewer for the question. We would like to first emphasize that we did not replace all LIF neurons in the network with DLIF neurons. Instead, we only substituted the LIF neurons in a subset of convolutional layers with DLIF neurons, resulting in an overall parameter increase of approximately 10%.
> >
> > Next, we clarify how the parameter counts of the linear LIF network and the DLIF network are computed, using the ResNet-34 as an example. For the standard LIF-based ResNet-34, the total number of learnable parameters is 21.8M. Most of these parameters come from convolutional layers. For a convolution layer with $C\_{in}$ input channels and $C\_{out}$ output channels, the LIF neuron in the output layer will linearly integrate the currents from the input channels. When introducing DLIF neurons, we consider the second-order interactions between input channels. In this case, the bilinear matrix is $C\_{in}\times C_{in}$ for each neuron in the output layer. Because we enforce 90% sparsity, the number of extra learnable parameters contributed by the bilinear term in a certain layer is:
> > $$
> > 0.1×C_{in}^2×C_{out}.
> > $$
> > In our implementation, we insert the bilinear DLIF interaction in the first two stages of ResNet-34. These stages include six convolution layers with channel sizes 64 and seven convolution layers with channel sizes 128. Thus the additional DLIF parameters are:
> >
> > $$
> > 0.1×(64^2×64×6+128^2×128×7)≈1.62M
> > $$
> > which corresponds to a 7.4% increase. This matches the parameter increase we report. We will update the manuscript to clearly specify which stages receive DLIF augmentation for different network architectures and tasks.

---

> > > ### Author Response · Authors · 2025-11-28
> > > **Response to Reviewer Uc9p (3/3)**
> > >
> > > **Question3:** The authors report the results of DLIF networks trained with OTTT. However, OTTT is an eligibility-trace-based online learning method designed for optimizing the linear weights. Its extension to bilinear weights is not trivial. Please explain how to use OTTT to optimize bilinear weights.
> > >
> > > **Response:** We thank the reviewer for the question.
> > > Suppose $u^{i,l+1}[t]$ is the voltage of the $i$th DLIF neuron in layer $l+1$ at time step $t$ in a feedforward network. When detaching the reset operation, the dynamics of  $u^{i,l+1}[t]$ is defined as
> > > $$
> > > u^{i,l+1}[t]=\lambda u^{i,l+1}[t-1]+(\mathbf{w}^l)^T\mathbf{s}^l[t-1]+(\mathbf{s}^l[t-1])^T\mathbf{K}^{l,i}\mathbf{s}^l[t-1]
> > > $$
> > >
> > > where $\lambda$ is the decay factor,  $\mathbf{s}^l$ is the spike trains of the $l$th layer neurons, $\mathbf{w}^{l}$ and $\mathbf{K}^{l,i}$ are the linear and bilinear weights from $l$th layer to the target neuron, respectively. When applying  the OTTT optimization, we use the same instantaneous loss as $L[t] = \frac{1}{T}\mathcal{L}(\mathbf{s}^N[t], \mathbf{y})$, where $T$ is the total time steps, $\mathcal{L}$ is the cross-entropy loss, $\mathbf{s}^N[t]$ is the spike at the last layer and $\mathbf{y}$ is the label.
> > >
> > > For the linear weight, the updating function is the same as in (Xiao et al., 2022):
> > >
> > > $$
> > > \frac{\partial L[t]}{\partial \mathbf{w}^l}=\mathbf{g}^{u^{l+1,i}}\mathbf{\hat{a}}^l[t]
> > > $$
> > >
> > > where $\mathbf{g}^{u^{l+1,i}}=(\frac{L[t]}{\mathbf{s}\_N[t]}\prod\_{i=N-1}^{l+1}\frac{\partial s[t]}{\partial s[t]}\frac{\partial s[t]}{\partial u})$ and $\mathbf{\hat{a}}\_l[t]=\sum\_{\tau\leq t}\lambda^{t-\tau}\mathbf{s}^l[\tau]$.
> > >
> > > For the bilinear weight, the updating function is defined in a similar way as:
> > > $$
> > > \frac{\partial L[t]}{\partial \mathbf{K}^{l,i}}=\mathbf{g}^{u^{l+1,i}}\mathbf{\hat{b}}^l[t]
> > > $$
> > > where $\mathbf{\hat{b}}^l[t]=\sum\_{\tau\leq t}\lambda^{t-\tau}(\mathbf{s}^l\[\tau])^T\mathbf{s}^l[\tau]$. In our implementation, we accumulate the gradients by T time steps and then update parameters.
> > >
> > > In this way, we use OTTT to optimize both the linear and bilinear weights in DLIF neurons.
> > >
> > > **Question4:** Typos: “biliear” in Related Work should be bilinear; “as a binary” in Page 3, Line 159 should be “as a binary matrix”.
> > >
> > > **Response:** We thank the reviewer for carefully identifying these typos. We have thoroughly checked the manuscript and corrected all of them in the revised version.
> > >
> > > **Question5:** Citation formatting: please use `\citep{}` instead of `(\cite{})`. The displayed text should appear as (xxx et al., 20xx) instead of (xxx et al., (20xx)).
> > >
> > > **Response:** We thank the reviewer for noting the citation formatting issue. We have carefully checked all references and replaced `(\cite{})` with `\citep{}` in the revised manuscript.
> > >
> > > **References**
> > >
> > > [1] Li, Songting, et al. "Dendritic computations captured by an effective point neuron model." Proceedings of the National Academy of Sciences 116.30 (2019): 15244-15252.
> > >
> > > [2] Zheng, Hanle, et al. "Temporal dendritic heterogeneity incorporated with spiking neural networks for learning multi-timescale dynamics." Nature Communications 15.1 (2024): 277.
> > >
> > > [3] Xiao, Mingqing, et al. "Online training through time for spiking neural networks." *Advances in neural information processing systems* 35 (2022): 20717-20730.
> > >
> > > [4] Zhou, Zhaokun, et al. "Spikformer: When Spiking Neural Network Meets Transformer." The Eleventh International Conference on Learning Representations. 2023.
> > >
> > > [5] Yao, Man, et al. "Attention spiking neural networks." *IEEE transactions on pattern analysis and machine intelligence* 45.8 (2023): 9393-9410.

---

### Official Review · Reviewer_9rV7 · 2025-10-31

**Soundness:** 2
**Presentation:** 3
**Contribution:** 2
**Rating:** 4
**Confidence:** 4

**Summary:**

This study proposed a new spiking neuron model called DLIF, which can conduct dendritic computations.  Experimental results show that this neural model achieves SOTA performance on five computer vision datasets

**Strengths:**

1. The proposed neuron model is biologically-inspired and the single neuron theorem is solid.
2. The experiments on XOR verified the theory.
3. Results are validated on several datasets

**Weaknesses:**

1. The energy metrics are estimated using numbers from a paper from a decade ago. Further justifications should be provided, and the energy benchmarking setting should be clearly presented, e.g., which GPUs are the target, how SNNs are set up and run on such GPUs, and how many GPU it required.
2. The paper compared many SNN algorithms and reproduced their results. It can be beneficial to provide codes (Assuming the ability to replace any neuronal models on any SNN algorithm is one of the main contributions of this paper), instead of just listing the experimental settings in the Supplementary materials for each SNN algorithm. It can make other researchers much easier to follow this study and develop a new spiking neuronal model on top of this study.

**Questions:**

1. Section 4 "Specifically, we set the sparsity level to 90%, a
choice that is both biologically inspired and empirically validated by an ablation study, as shown in
Section 4.4.". Please explain which dimension the sparsity is applied and how it is controlled before settingthe  sparsity level.

2. Is there any open-source plan for this study?

---

> ### Author Response · Authors · 2025-11-28
>
> **Weakness1**: The energy metrics are estimated using numbers from a paper from a decade ago. Further justifications should be provided, and the energy benchmarking setting should be clearly presented, e.g., which GPUs are the target, how SNNs are set up and run on such GPUs, and how many GPU it required.
>
> **Response**: We thank the reviewer for raising the question. Our energy estimation follows the neuromorphic-computing paradigm rather than GPU-based measurements. In this context, using theoretical per-operation energy costs from Horowitz (2014) is a well-established community standard. Both the paradigm and the metric are canonical in the SNN community and have been adopted in extensive prior studies (Kundu et al., 2021; Lemaire et al., 2022; Zhou et al., 2023; Yao et al., 2023; Hu et al., 2024; Yao et al., 2025).
>
>  We thank the reviewer again for pointing out the above issue and providing us an opportunity to clarify this rationale. We will cite these references in the revised manuscript.
>
> **Weakness2:** The paper compared many SNN algorithms and reproduced their results. It can be beneficial to provide codes (Assuming the ability to replace any neuronal models on any SNN algorithm is one of the main contributions of this paper), instead of just listing the experimental settings in the Supplementary materials for each SNN algorithm. It can make other researchers much easier to follow this study and develop a new spiking neuronal model on top of this study.
>
> **Response**:  We thank the reviewer for this suggestion. In compliance with the double-blind review policy, we have uploaded the code in the supplementary material. The official GitHub repository will be made publicly available upon acceptance.
>
> **Question1:** Section 4 "Specifically, we set the sparsity level to 90%, a choice that is both biologically inspired and empirically validated by an ablation study, as shown in Section 4.4.". Please explain which dimension the sparsity is applied and how it is controlled before setting the sparsity level.
>
> **Response**: We thank the reviewer for raising this question. In our formulation, each DLIF neuron is associated with a bilinear coefficient matrix $\mathbf{K}$ that integrates the presynaptic spike vector from the previous layer. Sparsity is applied to the entire bilinear coefficient matrix of each DLIF neuron. To control the sparsity level, we generate a fixed binary mask $\mathbf{M}$ before training: 90% of the entries are set to 0 and the remaining entries are set to 1. During training, the bilinear matrix is parameterized as $\mathbf{\tilde{K}}=\mathbf{K}\odot\mathbf{M}$, where $\odot$ is the Hadamard product. This ensures that the desired sparsity pattern is strictly preserved throughout optimization.
>
> We will clarify this implementation detail in the revised manuscript.
>
> **Question2:** Is there any open-source plan for this study?
>
> **Response**: We have uploaded the code in the supplementary material. The official GitHub repository will be made publicly available upon acceptance.
>
> **References**
>
> 1. Horowitz, Mark. "1.1 computing's energy problem (and what we can do about it)." *2014 IEEE international solid-state circuits conference digest of technical papers (ISSCC)*. IEEE, 2014.
> 2. Lemaire, Edgar, et al. "An analytical estimation of spiking neural networks energy efficiency." *International conference on neural information processing*. Cham: Springer International Publishing, 2022.
> 3. Yao, Xingting, et al. "SpiNeRF: Direct-trained spiking neural networks for efficient neural radiance field rendering." *Frontiers in Neuroscience* 19 (2025): 1593580.
> 4. Zhou, Zhaokun, et al. "Spikformer: When Spiking Neural Network Meets Transformer." The Eleventh International Conference on Learning Representations. 2023.
> 5. Yao, Man, et al. "Attention spiking neural networks." *IEEE transactions on pattern analysis and machine intelligence* 45.8 (2023): 9393-9410.
> 6. Kundu, Souvik, Massoud Pedram, and Peter A. Beerel. "Hire-snn: Harnessing the inherent robustness of energy-efficient deep spiking neural networks by training with crafted input noise." *Proceedings of the IEEE/CVF international conference on computer vision*. 2021.
> 7. Hu, Yifan, et al. "Advancing spiking neural networks toward deep residual learning." *IEEE transactions on neural networks and learning systems* 36.2 (2024): 2353-2367.

---

### Meta-Review · Area_Chair_c1U8 · 2026-01-06

**Summary:**

This paper proposes the Dendritic LIF (DLIF) neuron, which augments the standard LIF model with a bilinear dendritic integration term to capture input correlations beyond linear summation. The authors provide theoretical analysis at both single-neuron and network levels and demonstrate consistent empirical improvements across a wide range of static and neuromorphic vision benchmarks and SNN architectures. The work is biologically motivated and technically sound, with strong experimental coverage and generally clear presentation.

**Reviewer Concerns:**

The main concerns relate to the clarity and strength of the contribution relative to prior dendritic and multi-compartment neuron models, as well as the practical implications of the added bilinear term. Several reviewers questioned whether DLIF constitutes a fundamentally new neuron model or a scalable adaptation of existing dendritic formulations, and whether comparisons with recent dendritic SNN models were sufficiently comprehensive. The authors addressed many of these issues in the rebuttal with added experiments and clarifications.

**Reviewer Scores:**

Reviewer scores are mixed and skew slightly below the acceptance threshold. While one reviewer rated the paper as a clear accept and emphasized the strong empirical performance and breadth of evaluation, multiple reviewers rated the work as marginal reject, primarily due to concerns about novelty, comparison scope, and practical overhead.

---

### Decision · Program_Chairs · 2026-01-26

Accept (Poster)